# Learning from Suboptimal Data in Continuous Control via Auto-Regressive Soft Q-Network

**Jijia Liu** [1]  **Feng Gao** [1]  **Qingmin Liao** [1]  **Chao Yu** [1,2]  **Yu Wang** [1]

## Abstract

Reinforcement learning (RL) for continuous control often requires large amounts of online interaction data. Value-based RL methods can mitigate this burden by offering relatively high sample efficiency. Some studies further enhance sample efficiency by incorporating offline demonstration data to "kick-start" training, achieving promising results in continuous control. However, they typically compute the Q-function independently for each action dimension, neglecting interdependencies and making it harder to identify optimal actions when learning from suboptimal data, such as non-expert demonstration and online-collected data during the training process. To address these issues, we propose *Auto-Regressive Soft Q-learning* (ARSQ), a value-based RL algorithm that models Q-values in a coarse-to-fine, auto-regressive manner. First, ARSQ decomposes the continuous action space into discrete spaces in a coarse-to-fine hierarchy, enhancing sample efficiency for fine-grained continuous control tasks. Next, it auto-regressively predicts dimensional action advantages within each decision step, enabling more effective decision-making in continuous control tasks. We evaluate ARSQ on two continuous control benchmarks, RLBench and D4RL, integrating demonstration data into online training. On D4RL, which includes non-expert demonstrations, ARSQ achieves an average $1.62\times$ performance improvement over SOTA value-based baseline. On RLBench, which incorporates expert demonstrations, ARSQ surpasses various baselines, demonstrating its effectiveness in learning from suboptimal online-collected data.

## 1. Introduction

Deep reinforcement learning (RL) has demonstrated remarkable performance across various continuous control domains (Haarnoja et al., 2018; Schulman et al., 2017). However, these breakthroughs often come at the cost of extensive online interactions, which are required for effective convergence (Berner et al., 2019; Mnih et al., 2015). This reliance on large-scale exploration poses a major challenge in real-world applications, where data collection can be expensive, time-consuming, or even risky. To alleviate this burden, value-based RL methods, which directly approximate the Q-function rather than parameterizing a policy, have gained popularity due to their improved sample efficiency (Seyde et al., 2024; Tavakoli et al., 2021; Seyde et al., 2023) and have shown advances in continuous control tasks by discretizing each of the dimensions of continuous action spaces (Seo et al., 2024). Moreover, some studies integrate offline demonstration data into training to further accelerate early learning, reducing the dependence on purely online exploration (Ball et al., 2023). In this paper, we adopt this training paradigm to address continuous control using value-based RL, incorporating offline data into the online training process.

For value-based RL, the discretization scheme results in an exponentially large discrete action space, making RL training and exploration challenging. To mitigate this, existing value-based methods often estimate the Q-value for each action dimension independently (Metz et al., 2017; Seyde et al., 2023). However, this simplification comes with a limitation—it neglects interdependencies between action dimensions, potentially leading to suboptimal decision-making. When training data exhibits multiple modes, such as a mix of optimal and suboptimal demonstrations, independently estimating Q-values can bias action selection toward the most frequent behaviors rather than the truly optimal ones. This limitation is particularly pronounced in the early stages of learning, when the agent relies heavily on imperfect offline data and lacks sufficient online refinement.

Consider a simple one-step decision-making task with two-

[1]Tsinghua University, Beijing, China [2]Beijing Zhongguancun Academy, Beijing, China. Correspondence to: Chao Yu <zoeyuchao@gmail.com>, Yu Wang <yuwang@tsinghua.edu.cn>.

*Proceedings of the 42^{nd} International Conference on Machine Learning*, Vancouver, Canada. PMLR 267, 2025. Copyright 2025 by the author(s).

Project page is https://sites.google.com/view/ar-soft-q.

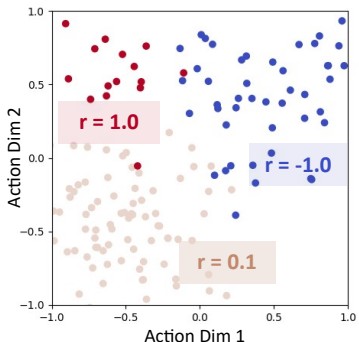
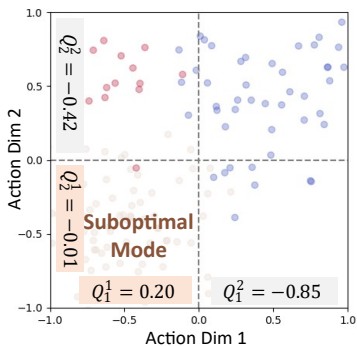
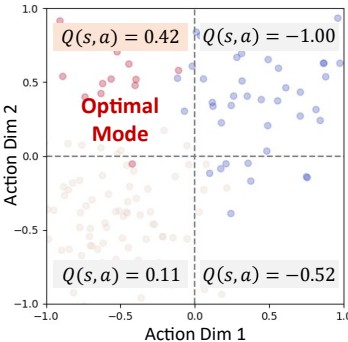

(a) An example dataset for a one-step decision-making environment.

(b) Q function given by independent action decomposition.

(c) Q function given by auto-regressive action decomposition (Ours).

*Figure 1.* A motivating example of how Q decomposition influences policy training, as detailed in Appendix C.1.

dimensional actions $(a_1, a_2) \in \mathcal{A} = [-1, 1]^2 \subset \mathbb{R}^2$, shown in Fig. 1, where an agent selects an action $(a_1, a_2)$ given state $s$ and receives a reward $r$ before the episode terminates. Suppose the training dataset consists of three distinct modes: one optimal mode with $r = 1$, two suboptimal modes with $r = 0.1$ and $r = -1$, with the latter occurring more frequently. If the suboptimal modes are more prevalent in the dataset, conventional Q-learning approaches that estimate action dimensions independently, i.e., $Q(s, a_i)$, could undervalue the optimal mode. This bias can hinder the correct identification and reinforcement of the optimal action mode, leading to slow convergence and degraded policy performance.

To address this issue, we propose *Auto-Regressive Soft Q-learning* (ARSQ), a novel approach that captures cross-dimensional dependencies in discretized high-dimensional action spaces. Instead of treating each dimension independently, ARSQ adopts an auto-regressive structure, sequentially estimating advantages for each action dimension conditioned on the previously selected dimensions. This allows the method to better model interdependencies, ensuring that correlated action dimensions are jointly optimized rather than selected in isolation. Additionally, ARSQ adopts a coarse-to-fine hierarchical discretization strategy inspired by CQN (Seo et al., 2024), further enhancing sample efficiency for fine-grained continuous control. We theoretically show that the original Q function can be expanded into an auto-regressive formulation with dimensional advantage estimation under the framework of soft Q-learning. Our approach integrates these insights into an auto-regressive soft Q-network, which is specifically designed for continuous control tasks.

To evaluate ARSQ, we conduct extensive experiments on the D4RL and RLBench continuous control benchmarks, challenging it against a variety of widely used reinforcement

learning and imitation learning baselines. Results indicate that ARSQ consistently surpasses these baselines, achieving up to $1.62\times$ performance over existing value-based RL when trained with suboptimal demonstrations on D4RL. Ablation studies further highlight the significance of ARSQ's key components, confirming its effectiveness in continuous control tasks.

Our contributions include:

- We extend Soft Q-learning framework to value-based reinforcement learning with dimensional advantage estimation.

- We propose the ARSQ algorithm to capture dependencies in action dimensions and enhance learning from suboptimal data.

- Through extensive experiments, we demonstrate that ARSQ can learn better policies when data suboptimality arises from either offline datasets or data collected online.

## 2. Related Works

**Value-based RL for Continuous Control.** Despite their inherently straightforward critic-only framework, value-based reinforcement learning (RL) algorithms have achieved notable success (Mnih et al., 2015; Silver et al., 2017; Schrittwieser et al., 2020; Seyde et al., 2024; Seo et al., 2024). Although these algorithms are primarily designed for discrete action spaces, recent efforts have sought to adapt them to continuous control by discretizing the continuous action space (Tavakoli et al., 2018; Seyde et al., 2023). However, the curse of dimensionality remains a significant challenge, as the number of discretization bins increases exponentially with the action dimension (Lillicrap, 2015). To address this issue, some studies have modified the Markov Decision

Process (MDP) of the environment, transforming it into a sequential decision-making problem along the action dimension (Metz et al., 2017; Chebotar et al., 2023). Other approaches treat each action dimension independently, generating the Q function separately for each dimension (Tavakoli et al., 2018; 2021; Seyde et al., 2023; 2024), akin to treating each action dimension as a multi-agent RL problem (Foerster et al., 2018; Yu et al., 2022). Recent research (Seo et al., 2024) has employed a coarse-to-fine discretization approach to improve sample efficiency. However, treating each action dimension independently may disrupt the correlation between different action dimensions, potentially diminishing performance in policy optimization. Some studies (Seo & Abbeel, 2024) have attempted to solve this issue through action sequence prediction. Our approach generates actions in an auto-regressive manner, considering the correlations between dimensions and improving policy learning, which is orthogonal to (Seo & Abbeel, 2024).

**Online RL with Offline Demonstration.**   Deep reinforcement learning often requires a large amount of online interactions to achieve convergence (Berner et al., 2019; Mnih et al., 2015). To address this challenge, many methods have been proposed that leverage offline demonstrations to guide online exploration and accelerate policy training (Rajeswaran et al., 2018; Ball et al., 2023). Some approaches involve performing offline RL pretraining before initiating online RL training (Lee et al., 2022; Nakamoto et al., 2023; LEI et al., 2024; Hu et al., 2024). However, these approaches often depend on expensive offline pretraining. To mitigate this, some works explore incorporating offline demonstration data directly into the training process. One strategy initializes the replay buffer with offline data (Hester et al., 2018; Ball et al., 2023), while another balances sampling between online and offline data to improve training stability (Zhang et al., 2023; Hansen et al., 2023). Additionally, certain methods explicitly introduce a behavior cloning loss to leverage high-quality demonstrations for better guidance (Rudner et al., 2021; Rajeswaran et al., 2018; Nair et al., 2018). In this work, we adopt the paradigm of integrating offline demonstrations into training to enhance sample efficiency in continuous control tasks. Specifically, we improve value-based RL by introducing an auto-regressive structure that sequentially estimates advantage for each action dimension. This design enables better handling of suboptimal data, whether from offline demonstrations or trajectories collected during training.

## 3. Preliminaries

### 3.1. Problem Formulation

In this paper, we consider the standard RL setting with the addition of a pre-collected dataset $\mathcal{D}$ for continuous control. The problem can be represented as MDP, defined

by the tuple $(\mathcal{S}, \mathcal{A}, \gamma, p, r, d_0)$. Here, $\mathcal{S}$ is the continuous state space, $\mathcal{A}$ is the continuous action space, $\gamma \in (0, 1)$ is the discount factor, $p(s' \mid s, a)$ is the transition dynamics, $r(s, a)$ is the reward function, and $d_0(s)$ is the distribution of the initial state. In addition to interacting with the environment online, we assume access to a pre-collected dataset $\mathcal{D} = \{(s_i, a_i, r_i, s_i')\}$, which can substantially reduce sample complexity and provide broader state-action coverage.

### 3.2. Soft Q Learning

To improve policy exploration, maximum entropy RL enhances the reward by adding an entropy term (Ziebart et al., 2008; Haarnoja et al., 2017; 2018), so the optimal policy seeks to maximize entropy at every state it visits. The objective is defined as

$$J(\pi) = \sum_{t=0}^{T} \mathbb{E}_{(\mathbf{s}_t, \mathbf{a}_t) \sim \rho_\pi} \left[ r(\mathbf{s}_t, \mathbf{a}_t) + \alpha \mathcal{H}(\pi(\cdot | \mathbf{s}_t)) \right] \quad (1)$$

where $\mathcal{H}$ is entropy, $T$ is the episode length and $\rho_\pi$ is the trajectory distribution induced by policy $\pi$. The temperature parameter $\alpha$ dictates how much importance is placed on the entropy term in comparison to the reward. Let the soft Q-function and soft value function defined as:

$$Q_{\text{soft}}^*(\mathbf{s}_t, \mathbf{a}_t) = r_t +$$
$$\mathbb{E}_{(\mathbf{s}_{t+1}, \dots) \sim \rho_\pi} \left[ \sum_{l=1}^{\infty} \gamma^l \left( r_{t+l} + \alpha H \left( \pi^*(\cdot | \mathbf{s}_{t+l}) \right) \right) \right] \quad (2)$$

$$V_{\text{soft}}^*(\mathbf{s}_t) = \alpha \log \int_{\mathcal{A}} \exp \left( \frac{1}{\alpha} Q_{\text{soft}}^*(\mathbf{s}_t, \mathbf{a}') \right) d\mathbf{a}' \quad (3)$$

Then the optimal policy for Eq. (1) is given by

$$\pi^*(\mathbf{a}_t | \mathbf{s}_t) = \exp \left( \frac{1}{\alpha} \left( Q_{\text{soft}}^*(\mathbf{s}_t, \mathbf{a}_t) - V_{\text{soft}}^*(\mathbf{s}_t) \right) \right) \quad (4)$$

Similar to the standard Q-function and value function, the Q-function can be connected to the value function at a future state using a soft Bellman equation.

$$Q_{\text{soft}}^*(\mathbf{s}_t, \mathbf{a}_t) = r_t + \gamma \mathbb{E}_{\mathbf{s}_{t+1} \sim p(\mathbf{s}_t, \mathbf{a}_t)} \left[ V_{\text{soft}}^*(\mathbf{s}_{t+1}) \right] \quad (5)$$

The proof can be found in (Ziebart et al., 2008; Haarnoja et al., 2017).

## 4. Method

In this section, we begin by discussing the process of discretizing multi-dimensional actions in a coarse-to-fine manner. Building on this, we extend the soft Q-learning theory with a focus on the dimensional soft advantage. Subsequently, we introduce our *Auto-Regressive Soft Q-learning* (ARSQ) algorithm, which is overviewed in Fig. 2.

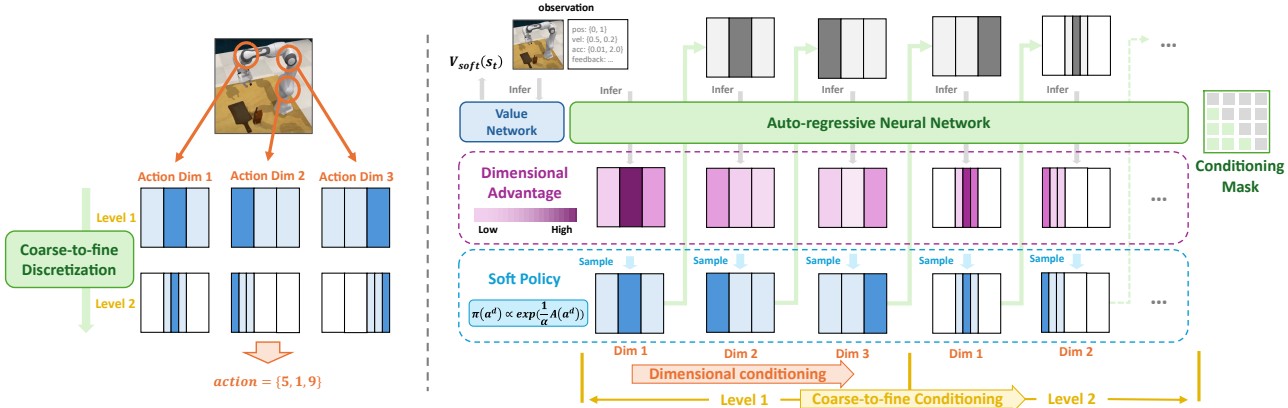

*Figure 2.* The ARSQ algorithm. The action space is discretized using a coarse-to-fine approach. By predicting dimensional soft advantages, ARSQ generates actions in an auto-regressive manner within a single decision-making step.

## 4.1. Coarse-to-fine Action Discretization

To apply Q-learning (Mnih et al., 2015) in a continuous domain, a straightforward approach is to discretize the action space (Tang & Agrawal, 2020; Seo et al., 2024). For a continuous action of $d$ dimensions $\mathbf{a}_c = (a_c^1, a_c^2, \ldots, a_c^d) \in \mathbb{R}^D$, the discretized action $\mathbf{a} = (a^1, a^2, \ldots, a^D)$ can be represented by

$$a^d = \arg\max_i |a_c^d - b_i| \qquad (6)$$

where $\mathbf{b} = (b_1, \ldots, b_B)$ are the centers of $B$ discretization intervals, or bins, which typically provide a uniform separation of the given action space. However, obtaining a finer separation of the action space necessitates a greater number of bins, thereby increasing the computational load when assessing the Q function for each discrete action bin.

To address this issue, we can apply a coarse-to-fine action discretization approach (Seo et al., 2024), similar to the method used in (Yan et al., 2015) for computer vision, as illustrated in Fig. 2. With $L$ levels and $B$ uniform separation bins at each level, the discrete action for dimension $d$ at level $l$ is expressed as:

$$a^{d,l} = \lfloor \frac{a^d - \sum_{i=1}^{l-1} B^{L-i} a^{d,i}}{B^{L-l}} \rfloor \qquad (7)$$

Here, $\lfloor \cdot \rfloor$ represents the floor function.

During inference, the policy generates discrete actions progressively through each level $(\mathbf{a}^{\langle \cdot \rangle, 1}, \mathbf{a}^{\langle \cdot \rangle, 2}, \cdots, \mathbf{a}^{\langle \cdot \rangle, L})$. These are then combined to produce the final discrete action.

## 4.2. Dimensional Soft Advantage for Policy Representation

Building on action discretization, we initially extend soft Q-learning to discrete spaces. The soft value function is

expressed as

$$V_{\text{soft}}^*(\mathbf{s}) = \alpha \log \sum_{\mathbf{a}' \in \mathcal{A}} \exp \left( \frac{1}{\alpha} Q_{\text{soft}}^*(\mathbf{s}, \mathbf{a}') \right) \qquad (8)$$

And we omit the subscript $t$ for $\mathbf{s}_t$ and $\mathbf{a}_t$ To further streamline the expression of the policy, we define the soft advantage.

**Definition 4.1** (Soft Advantage). The soft advantage of $\mathbf{a}$ at $\mathbf{s}$ is given by

$$A^*(\mathbf{s}, \mathbf{a}) = Q_{\text{soft}}^*(\mathbf{s}, \mathbf{a}) - V_{\text{soft}}^*(\mathbf{s}) \qquad (9)$$

Similar to the advantage in policy gradient-based RL algorithms, the soft advantage assesses how much taking action $\mathbf{a}$ at state $\mathbf{s}$ is beneficial. Thus, the optimal policy in Eq. (4) can be expressed as

$$\pi^*(\mathbf{a}|\mathbf{s}) = \exp \left( \frac{1}{\alpha} A^*(\mathbf{s}, \mathbf{a}) \right) \qquad (10)$$

Considering the multi-dimensional action space, it still remains necessary to use a neural network to output $B^{L \times D}$ Q values in the final layer, as per the DQN (Mnih et al., 2015).

However, outputting such a large number of Q values imposes a significant computational burden on the neural network. Inspired by auto-regression (Brown et al., 2020), we address this problem by make the policy $\pi$ generate action $\mathbf{a} = (a^1, a^2, \ldots, a^D)$ auto-regressively along the action dimensions.

For clarity, we treats discrete action discussed in Sec. 4.1 in one level. The multi-level coarse-to-fine discrete action can be considered as additional action dimensions, without compromising generalization. We first define the dimensional soft advantage to represent the auto-regressive policy at dimension $d$.

**Definition 4.2** (Dimensional Soft Advantage). The dimensional soft advantage of the action $a^d$ at state $\mathbf{s}$, considering the previous generated dimensional actions $\mathbf{a}^{-d} = (a^1, \cdots, a^{d-1})$, is expressed by

$$\pi(a^d|\mathbf{s}, \mathbf{a}^{-d}) \propto exp\left(\frac{1}{\alpha}A^d(\mathbf{s}, \mathbf{a}^{-d}, a^d))\right) \quad (11)$$

However, the dimensional soft advantage is not normalized for each action dimension $d$. We propose the following theorem to establish a connection between the dimensional soft advantage and the soft advantage.

**Theorem 4.3.** *If the dimensional soft advantage $A^d(\mathbf{s}, \mathbf{a}^{-d}, a^d)$ satisfies*

$$\sum_{a^d} \exp\left(\frac{1}{\alpha}A^d(\mathbf{s}, \mathbf{a}^{-d}, a^d)\right) = 1 \quad (12)$$

*for all dimension d, then the soft advantage can then be expressed as the summation of the dimensional soft advantages*

$$\sum_{d=1}^{D} A^d(\mathbf{s}, \mathbf{a}^{-d}, a^d) = A(\mathbf{s}, \mathbf{a}) \quad (13)$$

*Proof.* See Appendix A. $\quad\square$

Additionally, Eq. (12) shows that the exponential of the dimensional soft advantage represent a valid probability distribution. Using Eq. (11) together with Theorem 4.3, the dimensional soft advantage serves as a bridge between policy representation and Q prediction.

Since we do not introduce additional elements in policy optimization, the Q-iteration follows the same update rule as soft Q-learning. Based on Eq. (5), we have

$$V_{\text{soft}}(\mathbf{s}_t) + A(\mathbf{s}_t, \mathbf{a}_t) \leftarrow r_t + \gamma \mathbb{E}_{\mathbf{s}_{t+1} \sim p(s)}[V_{\text{soft}}(\mathbf{s}_{t+1})] \quad (14)$$

The maximum entropy policy described in Eq. (4) can be obtained by repeatedly applying Eq. (14) until it converges.

### 4.3. Auto-Regressive Soft Q-learning

Building on the theory outlined in Sec. 4.2, we introduce the *Auto-Regressive Soft Q-learning* (ARSQ) algorithm. The pseudo code for the ARSQ algorithm is presented in Algorithm 1. We will discuss the various design choices of ARSQ.

**Behavior Cloning Objective.** To leverage offline demonstration data during online training, we introduce an additional behavior cloning loss term. Following previous works (Hester et al., 2018; Seo et al., 2024), we encourage

---

**Algorithm 1** Auto-Regressive Soft Q Algorithm (ARSQ)

Initialize $\theta_{1,2}, \phi_{1,2}$ for $A^{\theta_i}$ and $V_{\text{soft}}^{\phi_i}$
Assign target parameters $\bar{\theta}_i, \bar{\phi}_i \leftarrow \theta_i, \phi_i$.
Offline dataset $\mathcal{D}$, replay buffer $\mathcal{R} \leftarrow \mathcal{D}$.
**for** each epoch **do**
  **for** each environment step **do**
    select $\mathbf{a}_t$ with $A_{\theta_1}$ and $A_{\theta_2}$ (10, 16)
    $\mathbf{s}_{t+1} \sim p(\mathbf{s}_{t+1}|\mathbf{s}_t, \mathbf{a}_t)$
    $\mathcal{R} \leftarrow \mathcal{R} \cup \{\mathbf{s}_t, \mathbf{a}_t, r_t, \mathbf{s}_{t+1}\}$
  **end for**
  **for** each gradient step **do**
    Sample mini-batch $b_D, b_R$ from $\mathcal{D}, \mathcal{R}$
    Calculate $\mathcal{L}_D = \mathcal{L}_{RL} + \beta\mathcal{L}_{BC}$ with $b_D$ (15, 18)
    Calculate $\mathcal{L}_R = \mathcal{L}_{RL}$ with $b_R$ (18)
    Update $m_{\theta_i}$ according to $\hat{\nabla}_{\theta_i}(\mathcal{L}_D + \mathcal{L}_R)$
    Update $V_{s,\phi_i}$ according to $\hat{\nabla}_{\phi_i}(\mathcal{L}_D + \mathcal{L}_R)$
    Update target networks $\bar{\theta}_i \leftarrow \rho\bar{\theta}_i + (1-\rho)\theta_i$ and
    $\bar{\phi}_i \leftarrow \rho\bar{\phi}_i + (1-\rho)\phi_i$.
  **end for**
**end for**

---

actions present in the offline dataset to be preferred over other actions. Specifically, we define the loss as

$$\begin{aligned}\mathcal{L}_{BC}^d = \sum_{a^d} \max(&A^{d,\theta_i}(\mathbf{s}, \mathbf{a}_e^{-d}, a^d) \\ &- A^{d,\theta_i}(\mathbf{s}, \mathbf{a}_e^{-d}, a_e^d), C_m)\end{aligned} \quad (15)$$

where $\mathbf{a}_e$ denotes the expert action observed in the offline dataset, and $C_m$ is a hyper-parameter controlling the margin. This objective encourages the soft advantages of expert actions to be at least $C_m$ higher than those of other actions.

**Policy Representation.** As discussed in Sec. 4.2, ARSQ predicts dimensional soft advantages, which function as both components of the Q function and policy representation. The network architecture is illustrated in Fig. 3. In practical design, the soft value $V_{\text{soft}}$ and the dimensional soft advantage $A^d$ are predicted using two separate neural networks. The advantage prediction network estimates the dimensional soft advantage for each action dimension, based on the partially generated action from previous dimensions, creating an auto-regressive sequence. In practical design, we use a globally-shared MLP in the advantage network, with separate heads to predict the dimensional soft advantages.

Another challenge is applying the constraint of the dimensional soft advantage as per Eq. (12). Here, we enforce a hard constraint by normalizing each output head through log-sum-exp subtraction, ensuring consistency across out-

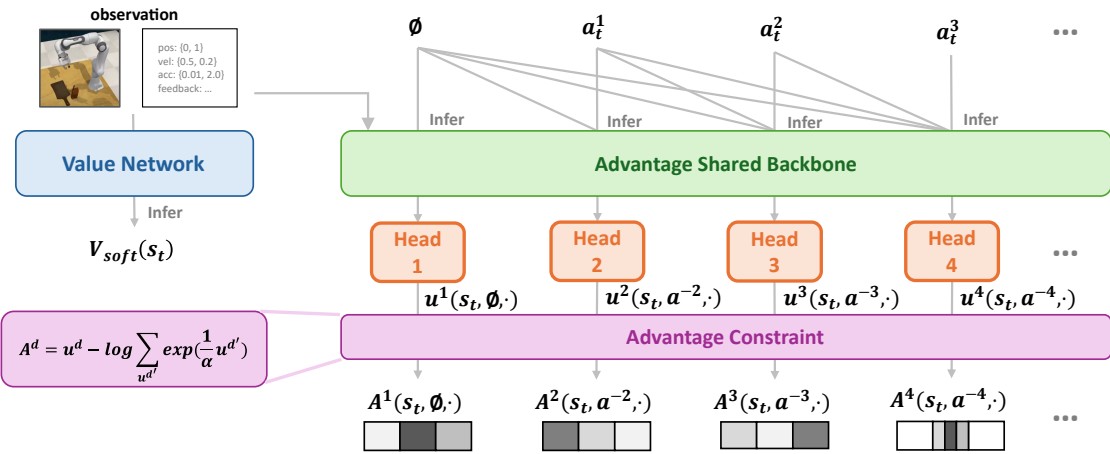

Figure 3. Network architecture of ARSQ. The soft value $V_{\text{soft}}$ and the dimensional soft advantage $A^d$ are predicted by two separate networks. The advantage network utilizes a shared backbone, and advantage constraints are applied to its output.

puts.

$$A^d(\mathbf{s}_t, \mathbf{a}^{-d}, a^d) = u^d(\mathbf{s}_t, \mathbf{a}^{-d}, a^d)$$
$$- \alpha log \sum_{a^{d'}} \exp\left(\frac{1}{\alpha} u^d(\mathbf{s}_t, \mathbf{a}^{-d}, a^{d'})\right) \quad (16)$$

where $u^d$ is the output of the $d$-th output head.

Furthermore, to stabilize training and address the over-estimation problem (Fujimoto et al., 2018; van Hasselt et al., 2016), we implement a double Q-learning approach with two separate value networks and their corresponding target networks. Specifically, we maintain two online value networks $V_{\text{soft}}^{\phi_1}$ and $V_{\text{soft}}^{\phi_2}$, along with their respective target networks $V_{\text{soft}}^{\overline{\phi}_1}$ and $V_{\text{soft}}^{\overline{\phi}_2}$. Each target network is updated as an exponential moving average (EMA) of its respective online network parameters. The value target is then computed by taking the minimum of the two target networks' predictions

$$\mathbf{y}_t = \gamma \mathbb{E}_{\mathbf{s}_{t+1} \sim p(s)} \left[ \min\left(V_{\text{soft}}^{\overline{\phi}_1}(\mathbf{s}_{t+1}), V_{\text{soft}}^{\overline{\phi}_2}(\mathbf{s}_{t+1})\right) \right] \quad (17)$$

Thus the resulting optimization objective becomes

$$\mathcal{L}_{RL} = \frac{1}{2} \left( V_{\text{soft}}^{\phi_i}(\mathbf{s}_t) + A^{\theta_i}(\mathbf{s}_t, \mathbf{a}_t) - \mathbf{y}_t \right)^2 \quad (18)$$

where $A^{\theta_i}$ is the soft advantage function parameterized by $\theta_i$.

**Auto-regressive Conditioning.** In Sec. 4.2, we explained the process of handling discrete action in one coarse-to-fine level. With multi-level coarse-to-fine action discretization, the auto-regressive conditioning encompasses two aspects. *Dimensional conditioning* refers to generating actions

for each dimension in an auto-regressive sequence, while *coarse-to-fine conditioning* involves generating actions for each dimension from coarse to fine. In practice, we implement coarse-to-fine conditioning prior to dimensional conditioning. Specifically, dimensional conditioning serves as the inner conditioning, while coarse-to-fine conditioning acts as the outer conditioning across levels. We explore swapping the order of conditioning in Sec. 5.4, and the results indicate that the current design better captures interdependencies between action dimensions.

## 5. Experiment

We design our experiments to investigate the following questions: (i) What is ARSQ's performance when the offline dataset is suboptimal? (ii) What is ARSQ's performance when online collected data is suboptimal? (iii) How do various design factors of ARSQ affect the performance?

**Benchmarks.** We evaluate our approach on two continuous control benchmarks: D4RL (Fu et al., 2020) and RLBench (James et al., 2020). Both domains provide access to online interaction data and a limited number of demonstrations, enabling us to assess the performance of ARSQ in diverse settings. We present representative results here due to limited space and leave full results in Appendix D.

**Baselines.** We use CQN (Seo et al., 2024), a state-of-the-art value-based RL method for continuous control, as our baseline. CQN employs a coarse-to-fine action selection strategy and independently predicts Q-values for each action dimension. Additionally, CQN trains using a combination of online training and offline demonstrations. Besides, we also include DrQ-v2 (Yarats et al., 2022), a renowned actor-critic algorithm designed for vision-based RL, along with its enhanced version, DrQ-v2+, as benchmarks. We also

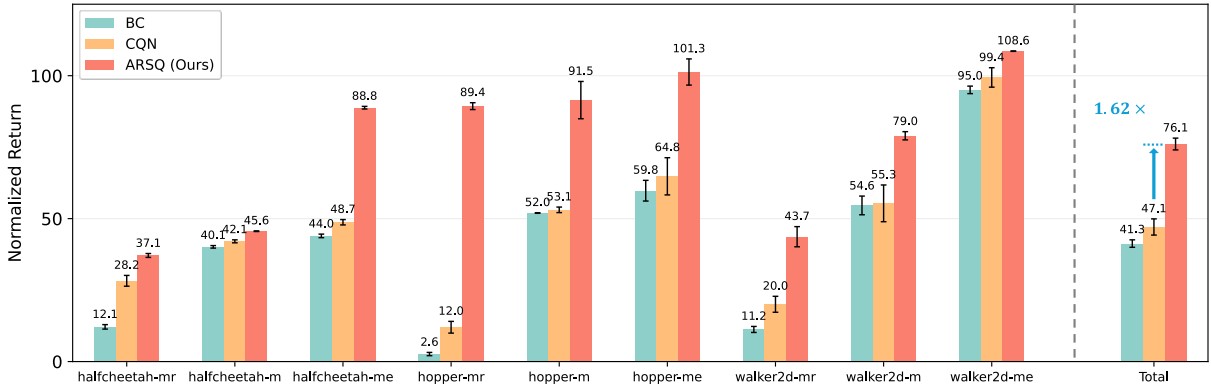

*Figure 4.* D4RL main results. *mr*, *m*, and *me* represent *medium-replay*, *medium*, and *medium-expert*, respectively.

feature ACT (Zhao et al., 2023) and a CQN-style behavior cloning (BC) policy among our baselines. Details about the baselines can be found in Appendix C.3.

### 5.1. Performance on D4RL

**Main Results.** To evaluate ARSQ's performance when the offline dataset is suboptimal, we consider three distinct locomotion tasks from the D4RL benchmark, each with three datasets of varying quality. The *medium* dataset is gathered using a medium-level policy, whereas the *medium-expert* dataset comprises a combination of medium-level and expert demonstrations. The *medium-replay* dataset includes data ranging from completely random to medium-level. The input to the model consists of state representations, while the output corresponds to torques applied at each hinge joint. A dense reward is provided to encourage completing the task, staying alive, and discourage vigorous actions that consume excessive energy.

We evaluate ARSQ, CQN (Seo et al., 2024), and BC in this setting. At the beginning of online training, the replay buffer for both ARSQ and CQN is initialized with an offline dataset, and online data is added as the training progresses. Additionally, both ARSQ and CQN incorporate the BC objective (Eq. (15)) towards offline dataset. The BC baseline is trained solely offline using the offline dataset with the BC objective. We report the converged performance of ARSQ, CQN and BC, averaged over three random seeds.

As shown in Fig. 4, ARSQ exhibits outstanding performance across all nine datasets, demonstrating its ability to effectively identify suboptimal actions and learn more efficiently from the available offline data. ARSQ surpasses CQN, particularly in the *medium-replay* and *medium-expert* datasets, where optimal data is not predominant, highlighting that ARSQ is less biased toward frequently observed suboptimal actions. Notably, both ARSQ and CQN outperform BC, indicating that conducting reinforcement learning online

enhances policy performance.

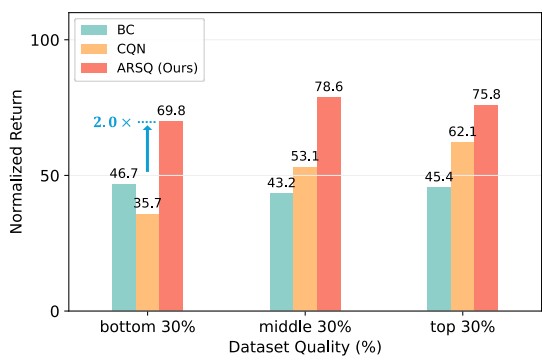

*Figure 5.* D4RL results on different demonstration quality averaged over 3 tasks, with each task containing 3 datasets respectively. We report the normalized return provided by D4RL.

**Analysis on Demonstration Quality.** To better investigate the influence of dataset quality, we rank trajectories by episode return for each dataset, and labeling the top 30%, middle 30%, and bottom 30% of the data as offline demonstrations. The behavior cloning objective is applied only to these offline demonstrations. We report the converged performance ARSQ, CQN and BC over three random seeds. As illustrated in Fig. 5, ARSQ consistently outperforms both CQN and BC across all three levels of demonstration quality. Notably, when using the bottom 30% of data as offline demonstrations, ARSQ achieves approximately 2.0× the final performance of CQN. In contrast, with the lowest demonstration quality, CQN performs slightly worse than BC, revealing CQN's sensitivity to demonstration quality, which negatively affects its online training. These results further validate the effectiveness of our method when using suboptimal offline datasets.

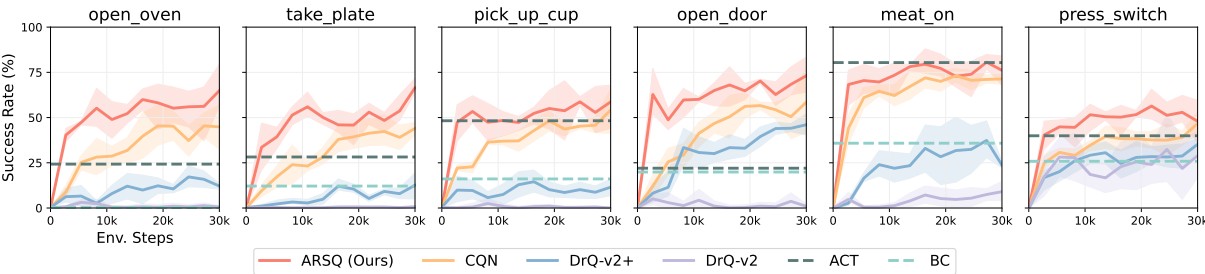

*Figure 6.* RLBench results on different tasks. Each experiment begins with 100 expert demonstrations, and all RL methods include a behavior cloning objective.

### 5.2. Performance on RLBench

To further evaluate ARSQ's performance, we focus on six tasks from RLBench (James et al., 2020). The agent receives input as RGB images and proprioceptive states and outputs the change in joint angles to control the robot arm. Unlike D4RL, the reward is sparse, offering a binary value (0 or 1) only at the final timestamp. Although each task is provided with 100 expert demonstrations, the agent might gather unsuccessful trajectories during its interaction with the environment. This setup allows us to examine the performance when the data collected online is suboptimal.

In this domain, we evaluate the performance of ARSQ, CQN, DrQ-v2+, DrQ-v2, ACT and BC. All reinforcement learning methods incorporate the behavior cloning objective (Eq. (15)) towards expert demonstrations and successful trajectories collected online. Results are averaged over three random seeds.

As shown in Fig. 6, ARSQ demonstrates superior performance compared to all other algorithms, highlighting its effectiveness in online learning with suboptimal collected data. Additionally, ARSQ exceeds ACT, highlighting the importance of reinforcement learning in online training.

### 5.3. Performance under Fully Offline Setting

To further examine the performance of ARSQ, we conduct experiments in a fully offline setting, where the algorithm learns solely from a predetermined dataset. We utilize nine locomotion datasets from D4RL, as outlined in Sec. 5.1. For offline RL methods, we employ CQL (Kumar et al., 2020), IQL (Kostrikov et al., 2022), TD3+BC (Fujimoto & Gu, 2021), Onestep RL (Brandfonbrener et al., 2021), and RvS-R (Emmons et al., 2022) as baselines. For offline imitation learning methods capable of handling suboptimal data, we use filtered BC (Chen et al., 2021; Emmons et al., 2022), Decision Transformer (Chen et al., 2021), and DWBC (Xu et al., 2022) as baselines. For DWBC, We adopt its best performance under "Setting 2" from its original paper (Xu et al., 2022), which mark top 5% trajectories as expert tra-

jectories based on total reward, and we re-evaluate DWBC under the same conditions on additional datasets.

The results are presented in Tab. 1. All data is sourced from the respective papers, with the reevaluated DWBC results marked by "*". Both ARSQ and re-evaluated DWBC are assessed using 10 trajectories over three random seeds. ARSQ demonstrates superior overall performance compared to the other baselines, indicating its capability to effectively manage suboptimal data under fully offline setting.

### 5.4. Ablation Studies

In this section, we evaluate the impact of key design factors in ARSQ: auto-regressive conditioning (Fig.2) and advantage prediction network (Fig.3).

**Ablation on Auto-regressive Conditioning.** We consider several variants of ARSQ on auto-regressive conditioning.

- *Swap*: We reverse the conditioning order, applying dimensional conditioning first, followed by coarse-to-fine conditioning.

- *w/o CF Cond.*: We remove the coarse-to-fine conditioning and output actions at multiple levels simultaneously.

- *w/o Dim Cond.*: We remove the dimensional conditioning and instead output all action dimensions simultaneously at each level.

- *w/o CF*: We replace the coarse-to-fine structure entirely by discretizing each action dimension into $B^L$ bins and then applying dimensional conditioning.

- *Plain*: We remove both the coarse-to-fine structure and dimensional conditioning.

We report results on *hopper-medium-expert* and *hopper-medium-replay* from D4RL, as well as *Open Oven* from RLBench, all evaluated across three random seeds. As depicted in Fig. 7, *Swap* demonstrates a slight decline in performance, underscoring the effectiveness of the current

*Table 1.* Performance under fully offline setting.

| Dataset | CQL | IQL | TD3+BC | Onestep RL | RvS-R | Filt. BC | DT | DWBC | ARSQ (Ours) |
|---|---|---|---|---|---|---|---|---|---|
| halfcheetah-m | 44.0 | 47.4 | 48.3 | **48.4** | 41.6 | 42.5 | 42.6 | *41.4 | 43.7 ± 0.6 |
| hopper-m | 58.5 | 66.3 | 59.3 | 59.6 | 60.2 | 56.9 | 67.6 | *56.0 | **99.2 ± 0.5** |
| walker2d-m | 72.5 | 78.3 | **83.7** | 81.8 | 71.7 | 75.0 | 74.0 | *72.3 | 81.2 ± 0.9 |
| halfcheetah-mr | **45.5** | 44.2 | 44.6 | 38.1 | 38.0 | 40.6 | 36.6 | 38.9 | 41.1 ± 0.1 |
| hopper-mr | 95.0 | 94.7 | 60.9 | **97.5** | 73.5 | 75.9 | 82.7 | 73.0 | 90.7 ± 4.4 |
| walker2d-mr | 77.2 | 73.9 | **81.8** | 49.5 | 60.6 | 62.5 | 66.6 | 59.8 | 74.0 ± 2.6 |
| halfcheetah-me | 91.6 | 86.7 | 90.7 | **93.4** | 92.2 | 92.9 | 86.8 | *93.1 | 92.4 ± 1.2 |
| hopper-me | 105.4 | 91.5 | 98.0 | 103.3 | 101.7 | **110.9** | 107.6 | *110.4 | **110.9 ± 1.0** |
| walker2d-me | 108.8 | 109.6 | 110.1 | **113.0** | 106.0 | 109.0 | 108.1 | *108.3 | 107.9 ± 0.3 |
| **Total** | 698.5 | 692.4 | 677.4 | 684.6 | 645.5 | 666.2 | 672.6 | 653.2 | **741.1** |

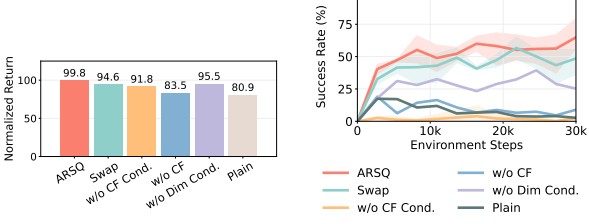

*Figure 7.* Ablation on auto-regressive conditioning in D4RL (left) and RLBench (right).

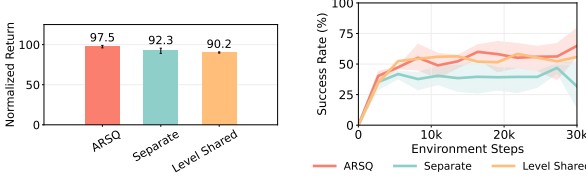

*Figure 8.* Ablation on shared backbone in D4RL (left) and RL-Bench (right).

conditioning order design. Additionally, removing any of the components degrades performance to varying degrees. When all components are removed, as in *Plain*, the performance is at its lowest, emphasizing the significance of dimensional and coarse-to-fine action generation.

**Ablation on Shared Backbone.** The advantage network of ARSQ utilizes a shared backbone to reduce the number of parameters and speed up the learning process. To assess the impact of this choice, we introduce two variants. The network architecture of these two variants can be found in Appendix C.3.

- *Separate*: We employ separate networks for each action dimension.
- *Level Shared*: We employ shared networks for each coarse-to-fine level.

We report results on *hopper-medium* from D4RL and *Open Oven* from RLBench over three random seeds. As shown in Fig. 8, the standard ARSQ consistently performs well in both environments. In contrast, using either a level-shared or separate backbone results in diminished performance. This demonstrates the effectiveness of the shared backbone design.

# 6. Conclusion

In this paper, we introduced *Auto-Regressive Soft Q-learning* (ARSQ), a novel value-based RL approach tailored for continuous control tasks with suboptimal data. ARSQ addresses the limitations of existing value-based methods by adopting an auto-regressive structure that sequentially estimates soft advantage for each action dimension, thereby capturing cross-dimensional dependencies. Through empirical evaluations, we show that ARSQ significantly surpasses existing methods, highlighting its effectiveness in learning from suboptimal data.

For future directions, an adaptive coarse-to-fine discretization can be used to balance control granularity with the overhead of additional bins. Another approach to explore is grouping unrelated dimensions to shorten the conditioning chain length, thereby speeding up computation.

# Acknowledgements

This work was supported by National Natural Science Foundation of China (No.62406159, 62325405), Postdoctoral Fellowship Program of CPSF under Grant Number (GZC20240830, 2024M761676), China Postdoctoral Science Special Foundation 2024T170496.

## Impact Statement

This paper presents work whose goal is to advance the field of Machine Learning. There are many potential societal consequences of our work, none which we feel must be specifically highlighted here.

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

## A. Proof of Theorem 4.3

First, we express the policy using conditional probability, and then replace it with Eq. (11).

$$
\begin{aligned}
\pi(\mathbf{a}|\mathbf{s}) &= \prod_{d=1}^{D} \pi(a^d|\mathbf{s}, \mathbf{a}^{-d}) \\
&= \prod_{d=1}^{D} \frac{exp\left(\frac{1}{\alpha}A^d(\mathbf{s}, \mathbf{a}^{-d}, a^d)\right)}{Z(\mathbf{s}, \mathbf{a}^{-d})} \\
&= \frac{\prod_{d=1}^{D} exp\left(\frac{1}{\alpha}A^d(\mathbf{s}, \mathbf{a}^{-d}, a^d)\right)}{\prod_{d=1}^{D} Z^d(\mathbf{s}, \mathbf{a}^{-d})} \\
&= \frac{exp\left(\frac{1}{\alpha}\sum_{d=1}^{D} A^d(\mathbf{s}, \mathbf{a}^{-d}, a^d)\right)}{\prod_{d=1}^{D} Z^d(\mathbf{s}, \mathbf{a}^{-d})}
\end{aligned}
\tag{19}
$$

We can then apply Eq. (12), resulting in

$$
\pi(\mathbf{a}|\mathbf{s}) = exp\left(\frac{1}{\alpha}\sum_{d=1}^{D} A^d(\mathbf{s}, \mathbf{a}^{-d}, a^d)\right)
\tag{20}
$$

Recall that the policy $\pi(\mathbf{a}|\mathbf{s})$ can be represented using the soft advantage as shown in Eq. (10). Therefore, we have

$$
\sum_{d=1}^{D} A^d(\mathbf{s}, \mathbf{a}^{-d}, a^d) = A(\mathbf{s}, \mathbf{a})
\tag{21}
$$

## B. Implementation Details

### B.1. Action Selection

As illustrated in Algorithm 1, the action selection process receives inputs from $A_{\theta_1}$ and $A_{\theta_2}$ and produces $\mathbf{a}_t$. Eq. (10) and Eq. (16) describe the action selection process utilizing a single soft advantage network. To leverage the benefits of a double network, we employ two advantage networks to generate more precise actions. This process is detailed in Algorithm 2.

---

**Algorithm 2** ARSQ Action Selection with Double Q Network

---

**Input:** parameter $\theta_{1,2}$ for $A^{\theta_i}$, state $\mathbf{s}_t$
**Output:** action $\mathbf{a}_t$
Initialize output action $\mathbf{a}_t = \emptyset$
**for** each action dimension $d$ **do**
    Compute $A^{d,\theta_i}(\mathbf{s}_t, \mathbf{a}_t, a^d)$ for each $a^d$ (16)
    Compute $A^d(a^d) = \min_i A^{d,\theta_i}(\mathbf{s}_t, \mathbf{a}_t, a^d)$
    Compute $\tilde{\pi}^d(a^d) = \exp\left(\frac{1}{\alpha}A^d(a^d)\right)$ (10)
    Normalize $\tilde{\pi}^d$ by $\pi^d(a^d) = \frac{\tilde{\pi}^d(a^d)}{\sum_{a^{d'}} \tilde{\pi}^d(a^{d'})}$
    Sample discrete action at dimension $d$ with $\pi^d(a^d)$
    Append action $\mathbf{a}_t = \mathbf{a}_t \cup \{a^d\}$
**end for**

---

### B.2. Variant of Behavior Cloning Objective

As discussed in Sec. 4.3, we incorporate an behavior cloning objective to effectively utilize offline demonstration data during online training, as defined in Eq. (15).

Following prior works (Kumar et al., 2020), we also employ a variant of this objective, expressed as:

$$
\mathcal{L}^d_{BC-v} = \max\left(\log \sum_{a^d \neq a^d_e} \exp\left(A^{d,\theta_i}(\mathbf{s}, \mathbf{a}^{-d}_e, a^d)\right) - A^{d,\theta_i}(\mathbf{s}, \mathbf{a}^{-d}_e, a^d_e), C_m\right)
\tag{22}
$$

where $a_e^d$ is the expert action and $C_m$ is a predefined margin constant.

We observe that this variant objective achieves better performance in scenarios where action modes are concentrated, such as in the *medium* and *medium-expert* series of datasets in D4RL. Consequently, we adopt this variant objective when working with such datasets.

### B.3. Network Architecture

In RLBench tasks, observations consist of a combination of RGB images and low-dimensional states. To compute the dimensional soft advantage for a given dimension, we first input the RGB images and low-dimensional states into a Convolutional Neural Network (CNN) (Li et al., 2022) encoder and a Multi-Layer Perceptron (MLP) (Haykin, 1998) encoder, respectively, to extract feature representations. These representations are then used to predict the soft value. Concurrently, the feature representations are combined with actions from previous dimensions and coarse-to-fine levels to create auto-regressive conditioning. An MLP-based shared backbone and output head are then utilized to determine the dimensional soft advantage for the given dimension.

In D4RL tasks, observations consist solely of low-dimensional states, and feature representations are derived directly from these states.

### B.4. Hyper-parameters

*Table 2.* Typical hyper-parameters of ARSQ in D4RL and RLBench.

| Hyper-parameter | D4RL | RLBench |
|---|---|---|
| Image resolution | / | $84 \times 84 \times 3$ |
| Image augmentation | / | RandomShift |
| Frame stack | 1 | 8 |
| CNN - Encoder | / | Conv (c=[32, 64, 128, 256], s=2, p=1) |
| Backbone | Linear (512, 512, 512) | Linear (512, 512, 512, bias=False) |
| Output Head Layers | 1 | 1 |
| Activation | Tanh | SiLU & LayerNorm |
| Coarse-to-fine Levels | 2 | 3 |
| Coarse-to-fine Bins | 7 | 5 |
| Batch Size | 512 | 512 |
| Optimizer | Adam | AdamW (weight decay = 0.1) |
| Learning Rate | 3e-4 | 5e-5 |
| Temperature Coefficient $\alpha$ | 0.01 | 0.001 |
| Target Critic Update Ratio ($\tau$) | 0.005 | 0.02 |
| BC Margin $C_m$ | -1 | -0.01 |
| Action Roll-out Network | Current | Target |

The hyperparameters of ARSQ are presented in Table 2. We provide the typical hyperparameters for ARSQ in D4RL (*hopper-medium*) and RLBench (*Open Oven*). In RLBench, ARSQ employs RandomShift (Yarats et al., 2022) for image augmentation. Additionally, ARSQ utilizes SiLU (Hendrycks & Gimpel, 2016) and LayerNorm (Ba, 2016) as activation functions in RLBench.

## C. Experiment Setup

### C.1. Motivating Example Setup

As introduced in Sec. 1 and illustrated in Fig. 1, we consider a motivating example to demonstrate the impact of Q decomposition on policy training. The dataset is depicted in Fig. 1a, with each point to be a data point in the dataset. The color of the data points indicates the reward of the data point. To illustrate the Q function of value-based RL algorithms, we first discretize the action space with 2 bins in each action dimension.

- Q function given by independent action decomposition is an example of DecQN (Seyde et al., 2023), as well as in CQN (Seo et al., 2024), which features just a single coarse-to-fine level. In this setting, we employ separate tabular Q functions, $Q(s, a_1)$ and $Q(s, a_2)$, for action dimension 1 and action dimension 2. The Q function is learned by gradient descent.

- For the Q function obtained through auto-regressive action decomposition, we employ both tabular soft advantage functions, $A^1(s, a_1)$ and $A^2(s, a_1, a_2)$ for action dimension 1 and action dimension 2, and a tabular soft value function $V_{\text{soft}}(s)$. The Q value reported in Fig. 1c is a sum of the soft value and the dimensional soft advantage of the corresponding dimensions, i.e., $Q(s, a_1, a_2) = V_{\text{soft}}(s) + A^1(s, a_1) + A^2(s, a_1, a_2)$. The soft advantage functions and the soft value function are simultaneously learned through gradient descent.

## C.2. Environment and Dataset

**D4RL Gym Environment.** D4RL (Fu et al., 2020) provides datasets for various tasks to evaluate the performance of reinforcement learning. In this context, we use 3 Gym Locomotion tasks and datasets from D4RL to assess the performance of ARSQ and other baselines. These tasks are illustrated in Fig. 9. The agent's observations include its states, such as the angle and velocity of each rotor. The agent's actions consist of torques applied between the robot's links, constrained within the range of $(-1, 1)$. The reward is dense, offering incentives for task completion and survival, while penalizing excessive energy-consuming actions.

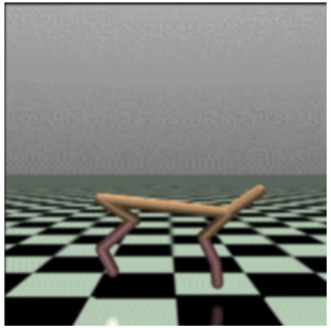 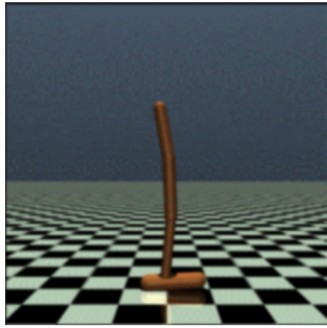 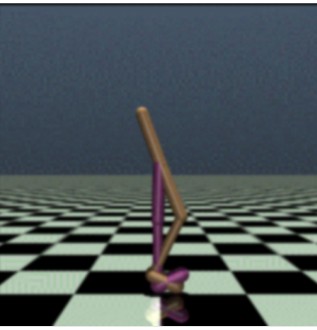

*Figure 9.* D4RL Gym tasks used in experiment.

**D4RL Dataset.** In D4RL, we use the *medium-replay*, *medium*, and *medium-expert* datasets for tasks involving *half-cheetah*, *hopper*, and *walker2d*. In Section 5.1, to examine the impact of dataset quality, we rank trajectories based on episode returns within these nine datasets. Specifically, we compute the total reward for each data chunk within each dataset. We then rank these data chunks and select the top, middle, and bottom $30\%$ accordingly. This is akin to rank trajectories but is easier to handle.

To better demonstrate the suboptimal nature of the datasets, we plot a histogram of the data chunk rewards, as shown in Fig. 10.

**RLBench Environment.** RLBench (James et al., 2020) serves as a benchmark and learning environment for robot control. We have selected 20 tasks from RLBench and present results for 6 of them in Sec. 5. An illustration of the environment can be seen in Fig. 11. The input consists of RGB images with a resolution of $84 \times 84$, captured from four camera angles: front, wrist, left-shoulder, and right-shoulder, along with a history of the past seven observations. The output specifies the change in joint angles at each time step, utilizing the delta JointPosition mode provided by RLBench. In our experiments, we use a binary sparse reward system (0 or 1), which is awarded only at the final timestamp of an episode to indicate task success.

## C.3. Baselines and Evaluation Details

**Main Results Baselines.** As mentioned in Sec. 5.1, within D4RL, we utilize the implementation from (Seo et al., 2024) and modify its CNN-based encoder to an MLP-based encoder as the CQN baseline. The BC baseline originates from CQN but operates with the RL learning objective turned off and without any online environment interaction.

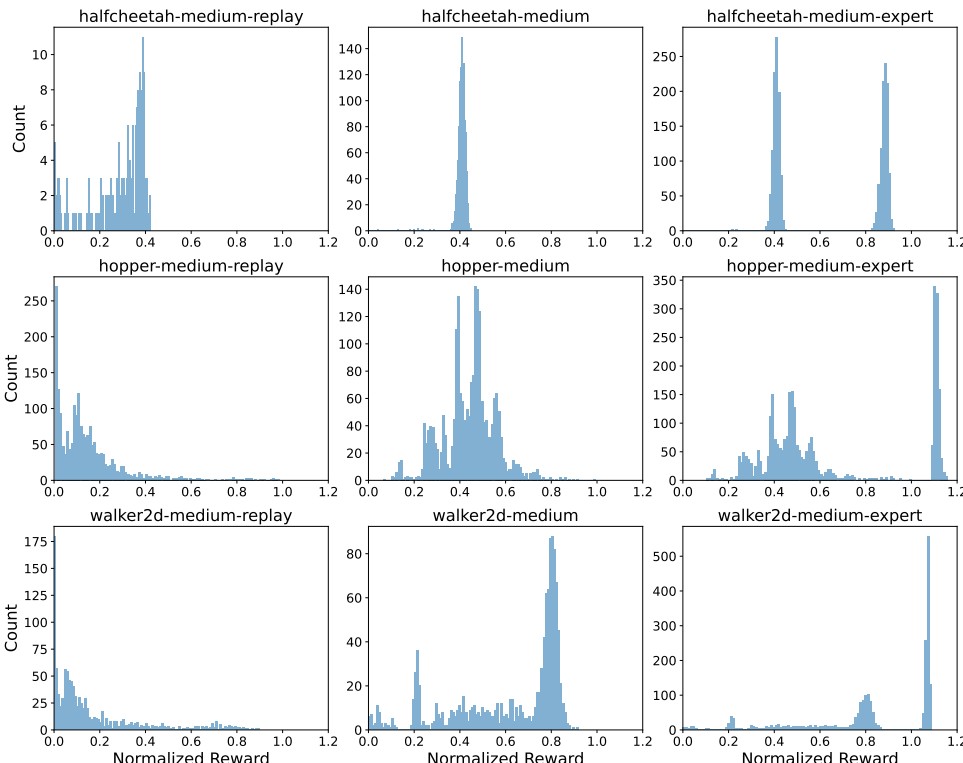

*Figure 10.* Histogram of reward in D4RL datasets.

In RLBench, we adopt DrQ-v2+, an optimized variant of DrQ-v2 proposed by (Seo et al., 2024), as our baseline. DrQ-v2+ incorporates several optimization strategies introduced in the CQN algorithm. Specifically, compared to DrQ-v2, DrQ-v2+ employs a distributional critic instead of a standard critic network, utilizes an exploration strategy with small Gaussian noise, and features optimized network architectures and hyperparameters tailored for RLBench tasks. These enhancements strengthen DrQ-v2+'s performance, making it a more robust baseline than DrQ-v2. Additionally, DrQ-v2+ has been open-sourced by (Seo et al., 2024).

**Ablation Study Baselines.** As mentioned in Sec. 5.4, we utilize the *Separate* and *Level Shared* backbone baselines for an ablation study to explore the effectiveness of the shared backbone in the advantage network. The network architectures of these two baselines are illustrated in Fig. 12 and Fig. 13.

## D. Additional Results

**Sensitivity of Temperature Coefficient** $\alpha$**.** Our methods are derived from Soft Q-learning, which aims to achieve a maximum-entropy policy. The temperature coefficient $\alpha$ in Eq. (1) affects the balance between maximizing policy entropy and the reward from the environment. We conducted experiments to examine how varying $\alpha$ impacts policy learning.

As shown in Fig. 14, a very high $\alpha$ results in reduced performance and unstable training, whereas a very low $\alpha$ also hampers policy improvement by restricting exploration.

**Training Curves of D4RL Main Results.** In Section 5.1, we discuss the converged performance of ARSQ, CQN, and BC. The training curves for each task are shown in Fig. 15. ARSQ converges after approximately 25,000 to 50,000 environment steps and generally outperforms the CQN and BC baselines across most tasks. This further demonstrates ARSQ's strength in managing suboptimal data.

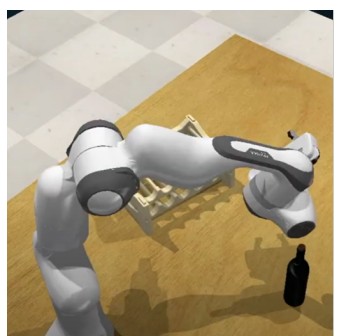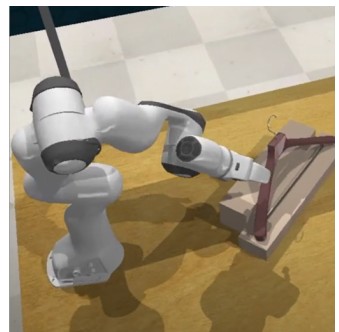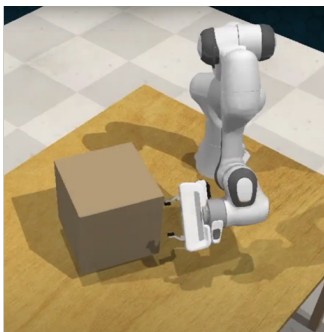

*Figure 11.* Example of RLBench tasks used in experiment.

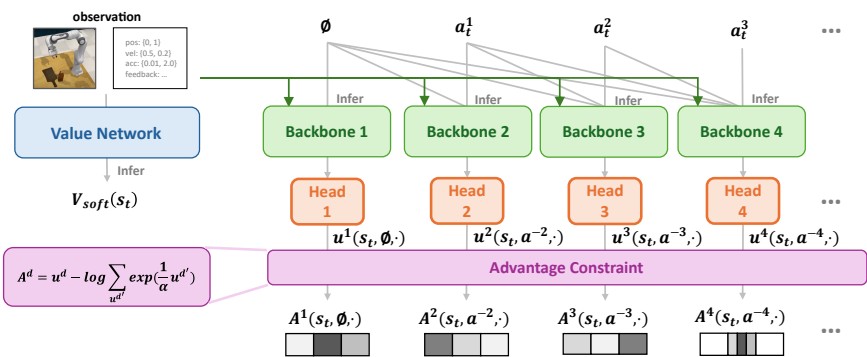

*Figure 12.* Network architecture of *Separate* backbone baseline in ablation study.

**D4RL Results per Task for Different Demonstration Quality.** In Sec. 5.1, we present the D4RL results, averaged over all 9 datasets, based on varying demonstration quality. The results for each task are illustrated in Fig. 16. ARSQ consistently outperforms the CQN and BC baselines in nearly every task, demonstrating its ability to maintain stable performance across datasets of varying quality.

**RLBench Results in All 20 Tasks.** In Sec. 5.2, we present results for six selected tasks from RLBench. The complete results for all 20 tasks are displayed in Fig. 17. These results indicate that ARSQ performs comparably or better across these tasks, showcasing its ability to learn effectively even when the data collected online is not optimal.

## E. Computational Cost Analysis

As discussed in Sec. 4.3, ARSQ generates actions in each dimension in an auto-regressive manner. To analyze the overhead, we conducted experiments on both D4RL (*hopper-medium*) and RLBench (*Open Oven*) tasks. The training and inference times for ARSQ and CQN were evaluated 1,000 times and averaged. These experiments were conducted on a single Nvidia RTX 3090 graphics card.

The results are shown in Fig. 3. ARSQ exhibits similar training times to CQN, due to the parallel optimization implemented and the batch training nature of the auto-regressive model. However, ARSQ experiences higher inference latency compared to CQN. We aim to address this issue by grouping the action dimensions and outputting the grouped dimensional actions auto-regressively, a solution we plan to explore in future work.

## F. Performance under Fully Online Setting

In addition to the main experimental results presented in Sec. 5.1, we assess the performance of ARSQ in a fully online setting. We compare the fully online reinforcement learning performance of ARSQ and CQN on the *hopper* task, using PPO(Schulman et al., 2017) as a baseline for comparison. The results are depicted in Fig. 18, with all experiments conducted

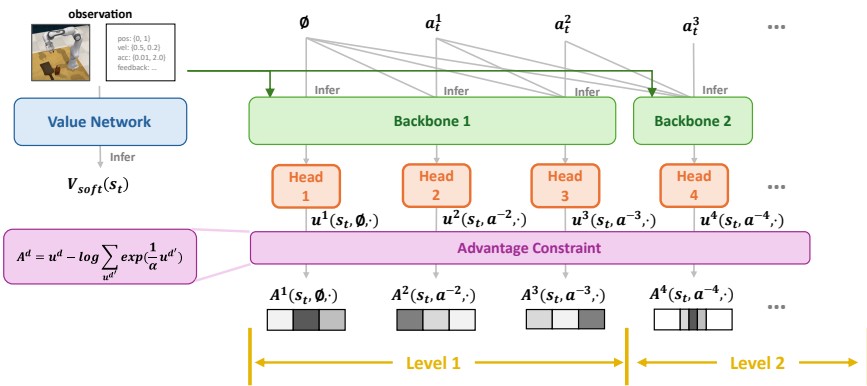

*Figure 13.* Network architecture of *Level Shared* backbone baseline in ablation study.

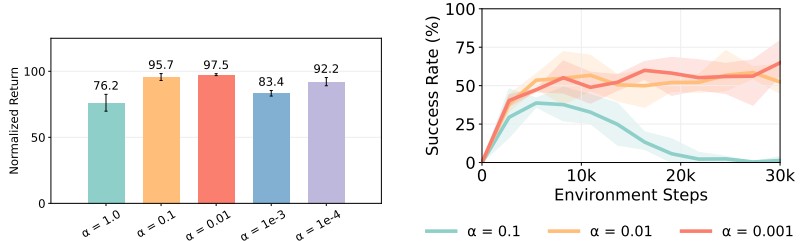

*Figure 14.* Sensitivity of temperature coefficient $\alpha$, evaluated on *hopper-medium* from D4RL and *Open Oven* from RLBench over three random seeds.

using three random seeds. We also illustrate the performance of both vanilla and offline ARSQ using the one of the *hopper* dataset, specifically *hopper-medium-replay*.

Online ARSQ achieves a similar converged performance to vanilla ARSQ, albeit requiring more environment steps. This highlights the importance of using offline datasets to enhance sample efficiency. Furthermore, ARSQ with online interaction achieves a higher final performance than in the offline setting, suggesting the necessity of online interaction to enhance policy performance. Additionally, online ARSQ demonstrates greater sample efficiency than CQN and PPO, underscoring its potential as a versatile reinforcement learning algorithm.

## G. Error Analysis of Q Prediction and Action Discretization

To further examine the error introduced by action discretization, as discussed in Sec. 4, we design a more complex case study similar to Fig. 1. We create a one-step environment featuring a two-dimensional action space $(a_1, a_2) \in \mathcal{A} = [-1, 1]^2 \subset \mathbb{R}^2$. The agent performs an action at the initial timestamp, receives a reward, and the episode concludes. The Q function's ground truth landscape, akin to the reward landscape, is illustrated in Fig. 19a. There is one optimal action mode, two sub-optimal action modes, and two negative action modes.

We uniformly sample 2,000 data points from the environment to form a dataset. This dataset is then used to train agents with independent Q decomposition for each action dimension, ARSQ without hierarchical coarse-to-fine action discretization, and the standard ARSQ. The resulting Q landscapes are displayed in Fig. 19, and Q prediction errors are displayed in Fig. 20. When independently decomposing Q for each action dimension, the agent learns a blurred Q landscape, complicating the identification of optimal actions. ARSQ without coarse-to-fine action discretization produces a Q landscape similar to the vanilla ARSQ but with more "glitches," likely because too many action bins make it difficult for the dataset to cover them comprehensively. This underscores the importance of coarse-to-fine action discretization.

Furthermore, we sample 1,000 data points in the proposed environment and calculate the Q prediction error against the ground truth for all three methods discussed, over three random seeds. The results are presented in Tab. 4. Independent Q

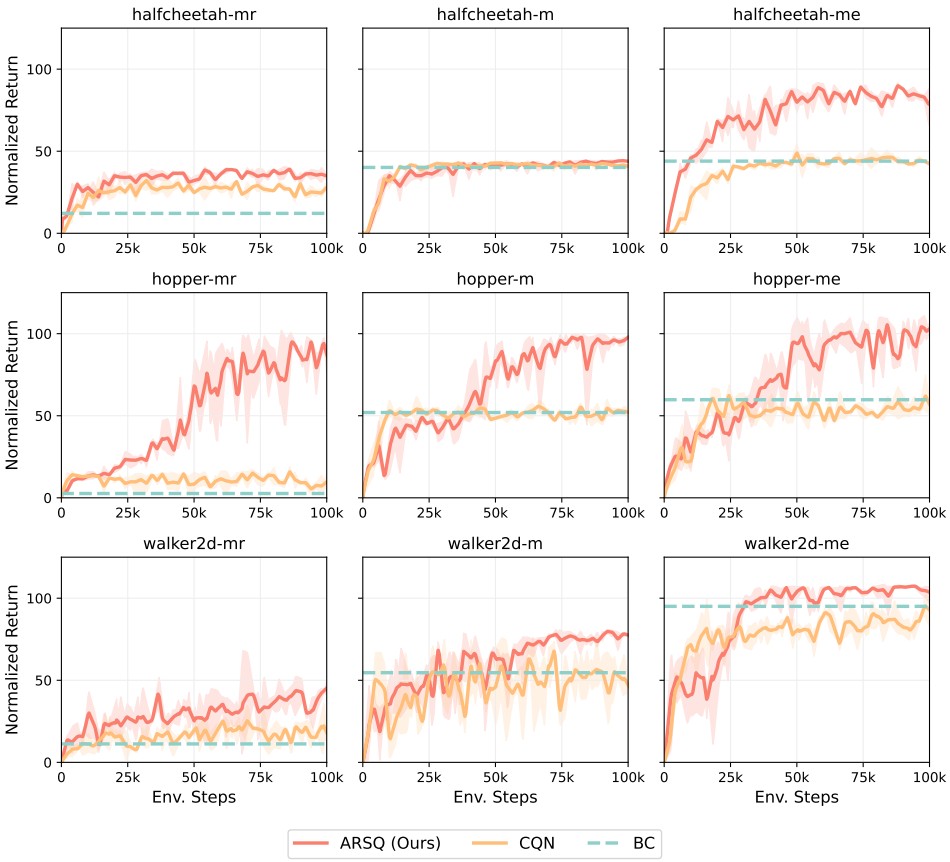

*Figure 15.* Training curves of D4RL main results, evaluated over three random seeds.

decomposition results in a significant error increase compared to ARSQ and continuous Q learning. Additionally, ARSQ without coarse-to-fine action discretization also results in higher Q error, further highlighting the necessity of our action discretization strategy.

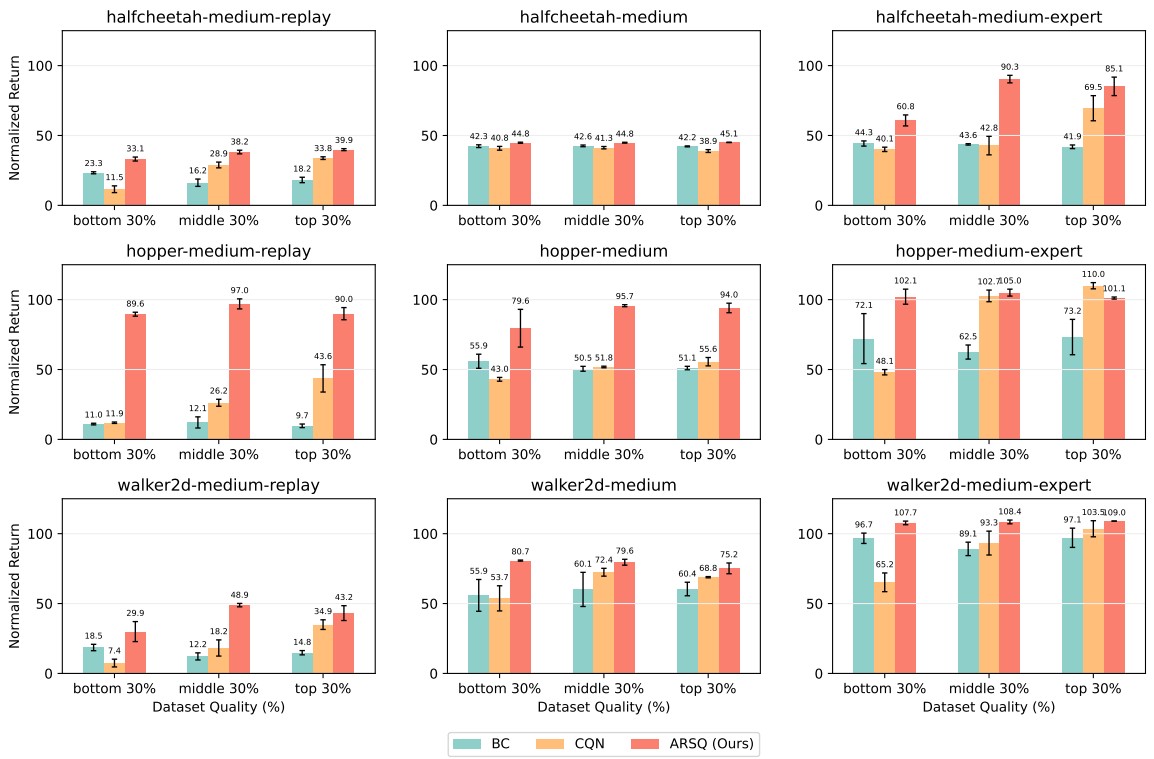

*Figure 16.* D4RL results per task on different demonstration quality, evaluated over three random seeds.

*Table 3.* Computational time in D4RL and RLBench (ms).

|                | D4RL | RLBench |
|----------------|------|---------|
| ARSQ Inference | 4.1  | 32.1    |
| ARSQ Training  | 12.2 | 290.5   |
| CQN Inference  | 2.6  | 6.9     |
| CQN Training   | 11.6 | 260.5   |

*Table 4.* Q prediction error with different action discretization strategy.

| Discretization Method      | Error           |
|----------------------------|-----------------|
| Independent Decomposition  | $17.57 \pm 0.67$ |
| ARSQ w/o Coarse-to-fine    | $0.50 \pm 0.21$  |
| ARSQ                       | $0.16 \pm 0.02$  |

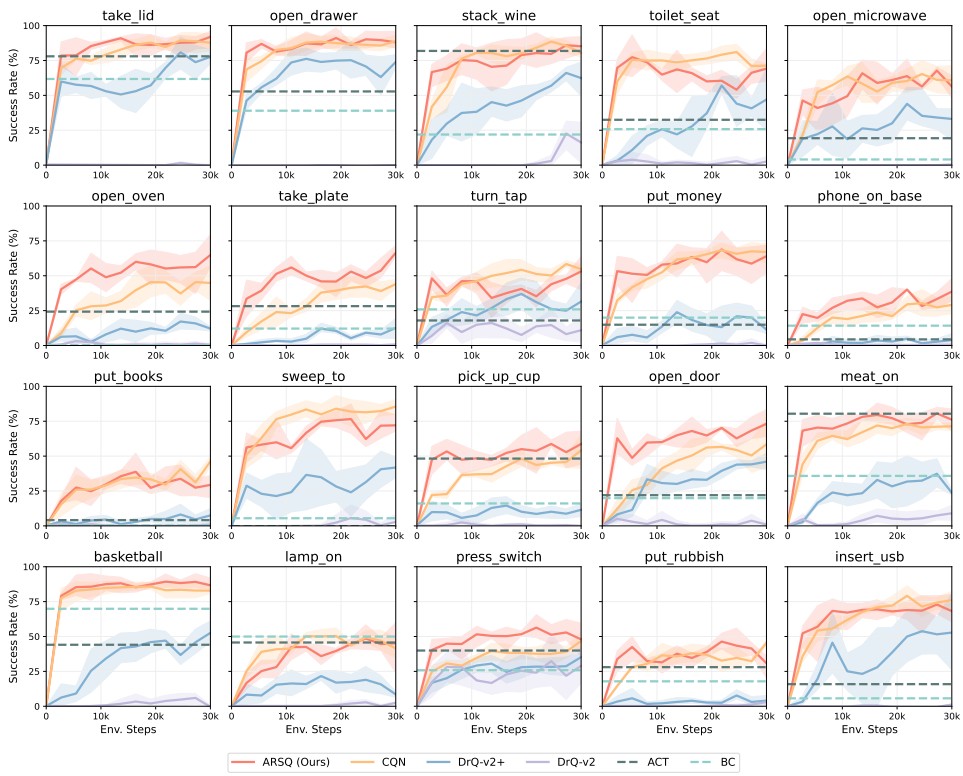

*Figure 17.* RLBench results in all 20 tasks.

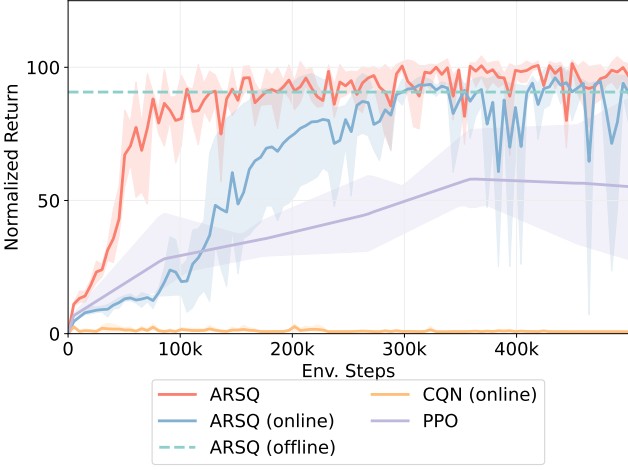

*Figure 18.* Performance under fully online settings, on *hopper* task or *hopper-medium-replay* dataset.

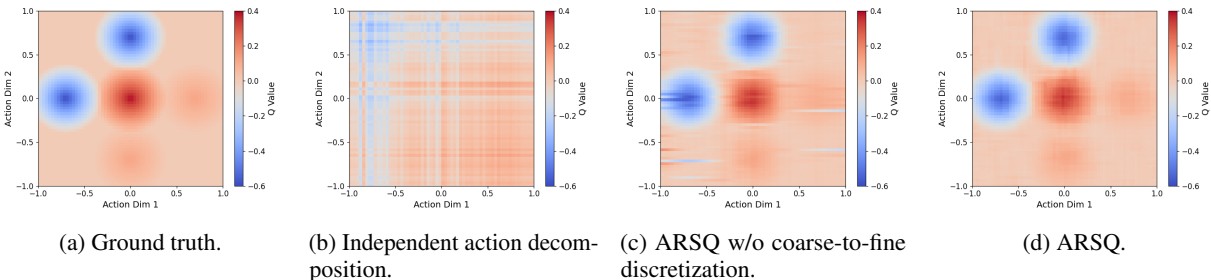

(a) Ground truth.   (b) Independent action decomposition.   (c) ARSQ w/o coarse-to-fine discretization.   (d) ARSQ.

*Figure 19.* Visualization of Q prediction with different action discretization strategy.

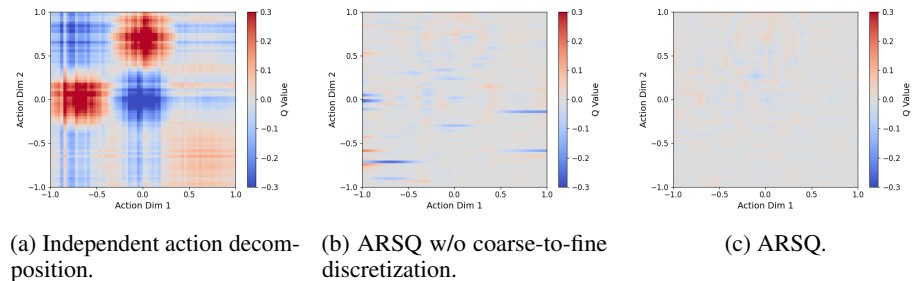

(a) Independent action decomposition.   (b) ARSQ w/o coarse-to-fine discretization.   (c) ARSQ.

*Figure 20.* Q prediction errors with different action discretization strategy.

