# OpenReview forum: "Learning from Suboptimal Data in Continuous Control via Auto-Regressive Soft Q-Network"
_ICML.cc/2025/Conference — ICML 2025 poster_

### Official Review · Reviewer_9ufZ · 2025-02-17

**Overall Recommendation:** 4

**Summary:**

This work introduces an algorithm for continuous control with discretized actions. Building upon coarse-to-fine Q learning, this algorithm further models advantages and policies in an autoregressive fashion, breaking the limiting assumption of independence between action dimensions. Update rules are derived from a soft variant of Q-learning, and are combined with a behavior cloning component. This method is evaluated on offline-to-online settings on D4RL and RLBench, displaying strong performance compares to purely offline baselines, or methods modeling action dimensions independently (Seo et al., 2024).

**Claims And Evidence:**

The main claim put forward by this work is in the empirical effectiveness of the proposed method. This claim is indeed supported by convincing evidence.

**Essential References Not Discussed:**

I am not aware of missing important references. My knowledge of related literature is however limited.

**Experimental Designs Or Analyses:**

The analyses are overall sounds, with minor issues with clarity as described in my questions.

**Methods And Evaluation Criteria:**

The evaluation criteria are consistent with the scope of this paper. However, the offline-to-online setup is rather restrictive. Can this ARSQ be applied, e.g., to fully online or fully offline settings?

**Other Comments Or Suggestions:**

None. The paper is well written, and I could not find any language issues nor typos.

**Other Strengths And Weaknesses:**

Strengths:
- The presentation is overall good, except for the aforementioned imprecisions in Section 4
- Empirical improvements appear to be significant.

Weaknesses:
- The theory is not precise, as described above (Eq. 3 and 10, Theorem 4.3).
- The experimental evaluation does not mention important information (see question below).

**Questions For Authors:**

1. Can you comment on my questions on Theoretical claims? Clarifications in this regard would help me confirm my score.
2. Why was the offline-to-online setup chosen? What prevents an evaluation in fully online or offline settings?
3. How many online steps are performed in Figure 4? Why are bar plots reported instead of training curves (as done in Figure 6)?
4. What tasks are evaluated in Figure 7? Is it an aggregate score over tasks?

## Update after rebuttal
Considering all of my comments were addressed, I have updated my score.

**Relation To Broader Scientific Literature:**

The proposed method proposes a solution to the important tradeoff between expressivity and tractability in continuous control with discretized actions. The autoregressive model proposed appears to solve the issues that arise when treating action dimensions independently, which is the standard approach in the literature, as far as I know.

**Theoretical Claims:**

Yes, the theoretical results are overall correct, although imprecise.

- I am skeptical about Equation 3 and 10: why can we assume that the exponentiated advantages are already normalized? Should the $=$ be replaced by a $\propto$? The rest of the derivations would still hold, as far as I can tell.

- Moreover, I find the theorem to be unnecessarily complicated. If I understand correctly, Eq. 12 is equivalent to assuming that exponentiated advantages represent a valid probability distribution. In this case, it's important to state this more clearly, and remark that this assumption is almost always violated.

---

> ### Author Rebuttal · Authors · 2025-04-01
>
> Thank you for your insightful review and valuable suggestions. We appreciate your careful reading and constructive
> feedback.
> Supplementary Material for our response is at
> [THIS LINK](https://anonymous.4open.science/r/icml25-Submission9509_2/fig_9ufZ.pdf).
>
> Below we address your specific questions and concerns one by one and provide clarifications and additional
> results as requested.
>
> ## 1. Clarifications on Theoretical Claims (Eq. 3 and Eq. 10)
>
> We are greatly thankful for your careful reading and for pointing out the ambiguity in the theoretical claims.
>
> Indeed, the exponential of advantages is inherently normalized due to its definition.
> In Soft Q Learning[1], the soft value function is defined using the soft Q function (Eq. 4)
>
> $$V^*_{\text{soft}}(s_t) = \alpha \log \int_{A} \exp ( \frac{1}{\alpha} Q^*_{\text{soft}}(s_t, a') ) da'$$
>
> Note that the soft value is a *softmax* of Q, and is *different* from the common concept of value
> function in RL. By construction, this directly implies:
>
> $$\int_{A} \exp ( \frac{1}{\alpha} ( Q^*_{\text{soft}}(s_t, a_t) - V^*_{\text{soft}}(s_t) )) da_t = 1$$
>
> which means that the exponentiated soft advantages are already normalized.
> This equality, crucial to our theoretical claims (Eq. 3 and Eq. 10 in our manuscript), has been rigorously
> proven in Theorem 1 of the original Soft Q-Learning paper [1].
>
> To address the reviewer’s valuable comment, We will explicitly clarify this point in the revised paper to strengthen the
> theoretical justification.
>
> [1] Haarnoja et al. (2017). Reinforcement learning with deep energy-based policies. ICML.
>
> ## 2. Clarification on Theorem 4.3
>
> Thank you for highlighting the complexity of Theorem 4.3.
>
> Theorem 4.3, which exactly means the exponential of the dimensional soft advantage represents a valid
> probability distribution, allows us to express the overall soft advantage function as the sum of the dimensional soft
> advantages.
> In this formulation, the dimensional soft advantage serves as a critical link between auto-regressive policy
> representation and Q-value prediction.
>
> We agree with the reviewer that the conditions of Theorem 4.3 are non-trivial and may not be satisfied naturally. To
> address this, we enforce a hard constraint by normalizing the output of the dimensional advantage prediction network.
> Specifically, we apply a log-sum-exp subtraction, as defined in Eq. (16). This normalization ensures that Theorem 4.3
> holds, thereby validating the correctness of all subsequent derivations.
>
> Additionally, we identified a typo in Eq. (16), missing temperature parameter $\alpha$. The corrected equation is:
>
> $$ A^d(\mathbf{s}_t, \mathbf{a}^{-d}, a^d) = u^d(\mathbf{s}_t, \mathbf{a}^{-d}, a^d) - \alpha log \sum _{a^{d'}} \exp ( \frac{1}{\alpha} u^d(\mathbf{s}_t, \mathbf{a}^{-d}, a^{d'}) )$$
>
> We will rearrange the theoretical claims and correct this typo accordingly in the revised manuscript.
>
> ## 3. Evaluation on Fully Offline and Fully Online Settings
>
> We agree with the reviewer that evaluating ARSQ in both fully offline and fully online settings is essential for
> completeness.
>
> **In the fully offline setting**, we have experiments in Tab. 1 in Supplementary Material, comparing ARSQ against
> representative offline RL and imitation learning methods. ARSQ demonstrates superior overall
> performance across tasks, highlighting its effectiveness on suboptimal offline data.
>
> **In the fully online setting**, we compare ARSQ against online CQN and PPO[2] in Fig. 1 in Supplementary Material.
> Online ARSQ achieves better sample efficiency than CQN and PPO, underscoring its potential as
> a general-purpose reinforcement learning algorithm.
> However, we observe that although online ARSQ eventually matches the converged performance of the standard ARSQ, it
> requires approximately $4\times$ more environment interactions
> to reach comparable performance, highlighting the importance of using offline datasets to enhance sample efficiency.
>
> [2] Schulman et al. (2017). Proximal policy optimization algorithms. arXiv:1707.06347.
>
> ## 4. Training Curves for D4RL Main Results (Fig. 4)
>
> Thank you for pointing out the absence of training curves. We have added detailed training curves for each task in Fig.
> 2 in Supplementary Material, clearly showing the number of online environment steps (approximately 25k–50k steps until
> convergence).
>
> We will also include these training curves in the revised manuscript to enhance clarity.
>
> ## 5. Tasks Evaluated in Figure 7
>
> Thank you for pointing out the lack of clarity regarding the tasks evaluated.
> Figure 7 reports results on two D4RL tasks (*hopper-medium-expert* and *hopper-medium-replay*), and one RLBench task
> (*Open Oven*).
>
> We will clarify this in the revised manuscript.
>
> We sincerely appreciate your constructive feedback, which has significantly improved the quality and clarity of our
> paper. We hope these clarifications and additional results address your concerns, and we are happy to further discuss
> any remaining questions.

---

### Official Review · Reviewer_L88h · 2025-03-13

**Overall Recommendation:** 3

**Summary:**

The paper proposes ARSQ, a value-based reinforcement learning method to improve learning from suboptimal data in continuous control tasks.  Previous methods estimate Q-values independently for each action dimension, neglecting their interdependencies, leading to biased action selection with mixed-quality data. ARSQ addresses this by modeling Q-values in an auto-regressive manner, A coarse-to-fine hierarchical discretization is porposed in improving efficiency in high-dimensional action spaces. Experiments on D4RL and RLBench benchmarks show that ARSQ achieves sota performance.

**Claims And Evidence:**

The paper offers good foundation for many of its claims, particularly through evaluations on standard benchmarks such as D4RL and RLBench. Still, tackling some remaining issues would further enhance the submission:

1. While ARSQ demonstrates improved performance when trained on suboptimal datasets, the nature of the suboptimality is not deeply analyzed.

2.The discretization method is discussed, but there is no direct comparison with alternative discretization strategies to prove its superiority.

**Essential References Not Discussed:**

[1] Jang, Y., Kim, G. H., Lee, J., Sohn, S., Kim, B., Lee, H., & Lee, M. (2023). SafeDICE: offline safe imitation learning with non-preferred demonstrations. Advances in Neural Information Processing Systems, 36, 74921-74951.

[2] An, G., Moon, S., Kim, J. H., & Song, H. O. (2021). Uncertainty-based offline reinforcement learning with diversified q-ensemble. Advances in neural information processing systems, 34, 7436-7447.

[3] Kostrikov, I., Nair, A., & Levine, S. (2021). Offline reinforcement learning with implicit q-learning. arXiv preprint arXiv:2110.06169.

**Experimental Designs Or Analyses:**

The experimental design in the paper is generally well-structured, leveraging D4RL and RLBench as benchmark datasets, comparing against strong baselines, and including ablation studies to isolate key design choices.

However, there are several areas where the soundness and validity of the experiments could be improved: There lacks comparison to offline RL methods (such as IQL, CQL, etc.), and RL/IL methods that handle suboptimal data (SafeDICE, EDAC). The reviewer believe a further comparison could enhance the soundness.

**Methods And Evaluation Criteria:**

The proposed ARSQ and evaluation criteria are largely appropriate for the problem of RL from suboptimal data in continuous control.

**Other Comments Or Suggestions:**

N/A.

**Other Strengths And Weaknesses:**

N/A.

**Questions For Authors:**

1 Theorem 4.3 assumes a normalization condition on the dimensional soft advantage function, Is there empirical validation or an approximation analysis to show its impact when the condition is violated?

2. The hierarchical discretization approach breaks down continuous action selection, but does it introduce bias in Q-learning updates? Would the method still converge to the optimal Q-function under function approximation errors?

3. Can you provide an error analysis of ARSQ’s predictions when trained on suboptimal data? How well does ARSQ generalize to unseen tasks or out-of-distribution environments?

**Relation To Broader Scientific Literature:**

The key contributions of the paper build upon several existing themes in reinforcement learning (RL), particularly in value-based RL for continuous control, handling suboptimal data, and hierarchical action representations.

**Theoretical Claims:**

The theoretical claims in the paper are generally well-motivated.

---

> ### Author Rebuttal · Authors · 2025-04-01
>
> We sincerely thank the reviewer for their thoughtful and constructive feedback.
> Supplementary Material for our response is at
> [THIS LINK](https://anonymous.4open.science/r/icml25-Submission9509_2/fig_L88h.pdf).
>
> Below, we address each point raised:
>
> ## 1. Analysis of the Nature of Suboptimality
>
> We agree that explicitly analyzing the nature of suboptimality is important, and provide two additional analyses to
> better illustrate suboptimality of datasets:
>
> - Trajectory Reward Analysis (Fig. 1 in Supplementary Material): We visualize histograms of trajectory rewards in D4RL
>   datasets, intuitively showing varying data quality.
>
> - Case Study (Fig. 2 in Supplementary Material): We introduce a simplified environment and compare ARSQ against action dimension independent Q decomposition methods.
>   The learned Q landscape demonstrates ARSQ's ability to learn accurate Q-values despite suboptimal data.
>
> ## 2. Comparison to Alternative Discretization Strategies
>
> We agree that ablations of alternative discretization strategies are important. In fact, we have provided some analyses
> along these lines in Section 5.3, where we evaluate several alternative discretization variants, including
> variants without hierarchical coarse-to-fine discretization (w/o CF Cond., w/o CF), variants that generate actions
> independently for each dimension (w/o Dim Cond., Plain), and a variant
> that swaps the conditioning order (Swap).
>
> The results show that ARSQ consistently outperforms all other variants, highlighting the effectiveness of our
> discretization strategy.
>
> ## 3. Comparison to Offline RL and Offline IL Methods
>
> We appreciate this valuable suggestion.
> In response, we have included additional comparisons between ARSQ and several popular offline RL and IL methods, in
> Table 1 of Supplementary Material.
> Regarding mentioned baselines:
>
> - **SafeDICE** primarily targets scenarios with labeled non-preferred trajectories. Thus, we adopt its preliminary
>   version, **DWBC**, which is more applicable to our setting.
> - **EDAC** is orthogonal and can be integrated to ARSQ in principle. However, due to time constraints, we leave this
>   integration for future work.
> - **IQL** and other offline RL/IL baselines are included in our revised paper.
>
> These comparisons further support the effectiveness of ARSQ in learning from suboptimal data.
>
> ## 4. Analysis of Theorem 4.3
>
> The normalization condition in Theorem 4.3 (Eq. 12) is indeed essential for the conclusion that the soft advantage can
> be decomposed into the summation of dimensional soft advantages (Eq. 13). Without this assumption, dimensional
> decomposition does not hold. To ensure this normalization constraint in practice, we
> apply a hard constraint during training via log-sum-exp subtraction (Eq. 16), ensuring consistency and theoretical
> validity.
>
> We also identified a typo in Eq. 16, where the temperature parameter $\alpha$ was inadvertently omitted. The corrected
> equation is:
>
> $$ A^d(\mathbf{s}_t, \mathbf{a}^{-d}, a^d) = u^d(\mathbf{s}_t, \mathbf{a}^{-d}, a^d) - \alpha log \sum _{a^{d'}} \exp ( \frac{1}{\alpha} u^d(\mathbf{s}_t, \mathbf{a}^{-d}, a^{d'}) )$$
>
> We will correct this typo in the revised manuscript.
>
> ## 5. Convergence to Optimal Q-function under Approximation Errors
>
> Our algorithm updates value and advantage functions via soft Bellman iteration (Eq. 14). Haarnoja et al. [1][2] have
> demonstrated that, in the absence of approximation errors, the optimal soft
> Bellman operator possesses a unique fixed point and is a contraction mapping in the infinity norm with contraction
> factor $\gamma < 1$. Thus, we expect our value iteration to converge close to the optimal solution if approximation
> errors remain bounded.
>
> [1] Haarnoja et al. (2017). Reinforcement learning with deep energy-based policies. ICML.
> [2] Haarnoja et al. (2018). Soft actor-critic: Off-policy maximum entropy deep reinforcement learning with a stochastic
> actor. ICML.
>
> ## 6. Hierarchical Discretization Bias and Error Analysis with Suboptimal Data
>
> We conducted an error analysis in a simplified 2D environment, in Fig. 2 & 3 and Tab. 2 in Supplementary Material.
>
> | Discretization | Q Prediction Error |
> |----------------|--------------------|
> | Independent    | 17.57 ± 0.67       |
> | ARSQ w/o CF    | 0.50 ± 0.21        |
> | ARSQ           | 0.16 ± 0.02        |
>
> The results demonstrate that both dimensional conditioning and hierarchical coarse-to-fine (CF) discretization
> significantly reduce Q-value prediction bias, further highlighting the necessity of our action discretization strategy.
>
> ## 7. Generalization to Unseen Tasks or Out-of-Distribution Environments
>
> We agree that analyzing ARSQ's generalization to unseen tasks or out-of-distribution environments is highly relevant.
> However, such analysis is beyond our current scope. We acknowledge this as an important future research direction.
>
> We thank the reviewer again for their valuable feedback, which significantly improves the clarity and rigor of our
> paper.

---

> > ### Comment · Reviewer_L88h · 2025-04-04
> >
> > Thank you for the detailed response. I will maintain my rating, with a positive inclination toward acceptance.

---

### Official Review · Reviewer_5ENF · 2025-03-14

**Overall Recommendation:** 4

**Summary:**

This paper studies the problem of reinforcement learning for continuous control with action discretization, specifically focusing on offline RL (D4RL benchmark) and RL from demonstrations (RLBench) as problem settings. A key limitation of prior work that leverages action discretization is the explosion in dimensionality that occurs if the continuous action space is jointly binned in a grid-like manner across all dimensions. Another option that prior work has explored is the use of separate action binning across each dimension, which avoids the aforementioned explosion but in turn makes an assumption about action dimension independence. This work proposes ARSQ, a method for coarse-to-fine hierarchical action discretization in an auto-regressive framework that overcomes both of these two issues by auto-regressively conditioning action dimension/coarse-to-fine sampling on the previously sampled ones (nicely illustrated in Figure 2). Experiments are conducted on D4RL and RLBench, and the proposed method outperforms CQN, an existing method for coarse-to-fine discretization, as well as recent methods for imitation learning (BC, ACT) and online RL (DrQ-v2).

## Post-rebuttal assessment

I appreciate the detailed response to my comments. The additional comparisons and clarifications address my concerns. I am raising my score from Weak Accept -> Accept under the assumption that these changes (along with those requested by my fellow reviewers) will be included in the camera-ready revision.

**Claims And Evidence:**

The main claims of the paper (that auto-regressive conditioning is beneficial, and that ARSQ outperforms existing methods on competitive benchmarks) are supported by empirical evidence. I believe that the proposed method is well motivated based on observed limitations in prior work, and the illustrative example in Figure 1 is helpful for understanding the problem. Ablations support the claim that the specific formulation of auto-regressive conditioning used in the proposed method is beneficial and favorable over a number of alternatives (e.g. conditioning only on dimensions or coarse-to-fine levels).

**Essential References Not Discussed:**

I am not aware of any works that are currently missing from the list of references, but it is possible that I missed some.

**Experimental Designs Or Analyses:**

See my previous comments in the "methods and evaluation criteria" field.

**Methods And Evaluation Criteria:**

The experimental setup is appropriate for evaluation of the method in question. D4RL and RLBench are competitive, commonly used environments in related literature, and the two benchmarks span both low-dimensional state representations as well as RGB images as inputs, and cover both pure offline RL and RLfD. Results are mostly convincing.

I presently have three concerns regarding the evaluation:

* There are few baseline results included for D4RL. I find this somewhat surprising given that it is a well established benchmark with lots of benchmark results readily available in prior literature. I would recommend that the authors include a few more baseline comparisons, e.g. IQL [1] and CQL [2]. I understand that this submission focuses on action discretization and that the aforementioned methods do not, but it would be helpful to have a set of established results on this benchmark for comparison. I believe that numbers can be extracted directly from the respective papers.

* Number of seeds is not specified, except for L350 referencing D4RL. It would be greatly appreciated if the authors could clearly detail the number of seeds used for each experiment/figure.

* Given that DrQ-v2+ is not a well established method but rather proposed as a baseline in prior work CQN, it would be helpful to include a description of this method in the submission (to make it self-contained) as well as a (potentially brief) comment on how the baseline was obtained/implemented (presumably on top of the public implementation of DrQ-v2).

[1] Kostrikov et al., Offline Reinforcement Learning with Implicit Q-Learning, https://arxiv.org/abs/2110.06169 (2021)

[2] Kumar et al., Conservative Q-Learning for Offline Reinforcement Learning, https://arxiv.org/abs/2006.04779 (2020)

**Other Comments Or Suggestions:**

It is clear from Eq. 17-18 how the value target is computed, but the couple of lines preceding that are a bit ambiguous. Maybe the authors can consider rephrasing it to make it explicit that they train two value networks and use two target networks that (presumably) are exponential moving averages of the two online value networks + that the value target then is computed as the minimum of the two targets. This is standard practice in the field but may not be obvious to uninitiated readers.

Also minor, but some of the figures are rather small and/or have small fonts which makes them difficult to read. I suggest that the authors revisit the formatting of the paper to mitigate that.

**Other Strengths And Weaknesses:**

In summary

**Strengths:** The paper is generally well written and easy to follow. The illustrations are helpful for understanding the problem setting and proposed method. The experimental setup appears solid (aside from the few concerns already mentioned), and empirical performance gains are substantial. I have no concerns regarding the originality of the work. I believe that the contributions will be of interest to the ICML community.

**Weaknesses:** I have a few concerns regarding the experimental evaluation, namely lack of (1) baseline results for D4RL, (2) clarity regarding number of seeds, and (3) details regarding the DrQ-v2+ baseline. I believe that these issues, while important, can easily be addressed during the rebuttal period.

**Questions For Authors:**

I would like the authors to address my previous comments regarding the experimental evaluation / weaknesses. Details about baselines such as DrQ-v2 and DrQ-v2+ could potentially be added to the appendices.

**Relation To Broader Scientific Literature:**

The work is generally well positioned wrt prior literature on the topic of action discretization.

**Theoretical Claims:**

I have not checked the theoretical claims in detail, but did not encounter any glaring issues while reading through the paper.

---

> ### Author Rebuttal · Authors · 2025-04-01
>
> We sincerely thank the reviewer for their detailed and constructive feedback. We appreciate the positive remarks about
> the motivation, clarity, originality, and empirical results of our work.
>
> Below, we respond to the reviewer’s concerns point-by-point:
>
> ## 1. Additional Baseline Results on D4RL
>
> We agree with the reviewer that additional baseline comparisons on the well-established D4RL benchmark would strengthen
> our evaluation.
> Following the reviewer’s suggestion, we have included results from several offline RL methods (CQL[1], IQL[2],
> TD3+BC[3], Onestep RL[4], RvS-R[5]) and offline imitation methods (Filtered BC[5], Decision Transformer(DT)[6],
> DWBC[7]).
>
> We have included the results in Table 1
> of [Supplementary Material](https://anonymous.4open.science/r/icml25-Submission9509_2/fig_5ENF.pdf).
> All data are sourced from the respective papers, and we re-evaluate DWBC with extending datasets (marked by "*").
> These results are summarized below for convenience:
>
> | Dataset        | CQL    | IQL   | TD3+BC | Onestep RL | RvS-R | Filt. BC | DT    | DWBC    | **ARSQ (Ours)** |
> |----------------|--------|-------|--------|------------|-------|----------|-------|---------|-----------------|
> | halfcheetah-m  | 44.0   | 47.4  | 48.3   | *48.4*     | 41.6  | 42.5     | 42.6  | \*41.4  | 43.7 ± 0.6      |
> | hopper-m       | 58.5   | 66.3  | 59.3   | 59.6       | 60.2  | 56.9     | 67.6  | \*56.0  | *99.2 ± 0.5*    |
> | walker2d-m     | 72.5   | 78.3  | *83.7* | 81.8       | 71.7  | 75.0     | 74.0  | \*72.3  | 81.2 ± 0.9      |
> | halfcheetah-mr | *45.5* | 44.2  | 44.6   | 38.1       | 38.0  | 40.6     | 36.6  | 38.9    | 41.1 ± 0.1      |
> | hopper-mr      | 95.0   | 94.7  | 60.9   | *97.5*     | 73.5  | 75.9     | 82.7  | 73.0    | 90.7 ± 4.4      |
> | walker2d-mr    | 77.2   | 73.9  | *81.8* | 49.5       | 60.6  | 62.5     | 66.6  | 59.8    | 74.0 ± 2.6      |
> | halfcheetah-me | 91.6   | 86.7  | 90.7   | *93.4*     | 92.2  | 92.9     | 86.8  | \*93.1  | 92.4 ± 1.2      |
> | hopper-me      | 105.4  | 91.5  | 98.0   | 103.3      | 101.7 | *110.9*  | 107.6 | \*110.4 | *110.9 ± 1.0*   |
> | walker2d-me    | 108.8  | 109.6 | 110.1  | *113.0*    | 106.0 | 109.0    | 108.1 | \*108.3 | 107.9 ± 0.3     |
> | Total          | 698.5  | 692.4 | 677.4  | 684.6      | 645.5 | 666.2    | 672.6 | 653.2   | **741.1**       |
>
> The above results demonstrate that ARSQ achieves competitive performance with existing offline RL/IL
> algorithms, further demonstrating its potential as a versatile RL algorithm.
>
> [1] Kumar et al. (2020). Conservative q-learning for offline reinforcement learning. NeurIPS 33.
> [2] Kostrikov et al. (2021). Offline reinforcement learning with implicit q-learning. arXiv:2110.06169.
> [3] Fujimoto et al. (2021). A minimalist approach to offline reinforcement learning. NeurIPS 34.
> [4] Brandfonbrener et al. (2021). Offline rl without off-policy evaluation. NeurIPS 34.
> [5] Emmons et al. (2021). Rvs: What is essential for offline rl via supervised learning? arXiv:2112.10751.
> [6] Chen et al. (2021). Decision transformer: Reinforcement learning via sequence modeling. NeurIPS 34.
> [7] Xu et al. (2022). Discriminator-weighted offline imitation learning from suboptimal demonstrations. ICML.
>
> ## 2. Clarification on Number of Seeds
>
> We apologize for the previous lack of clarity regarding the number of random seeds used.
> To clarify, all experiments were conducted using three different random seeds.
>
> We will ensure this information is clearly stated in the revised version of the paper.
>
> ## 3. Details on the DrQ-v2+ Baseline
>
> We thank the reviewer for pointing out this omission.
> DrQ-v2+ is an enhanced variant of DrQ-v2, proposed and open-sourced by Seo et al[8].
> It incorporates several key improvements, including a distributional critic, an exploration strategy using small
> Gaussian noise, and optimized hyperparameters.
> These enhancements make DrQ-v2+ a more competitive baseline compared to the original DrQ-v2.
>
> We will include a detailed description of the DrQ-v2+ baseline in the revised paper.
>
> [8] Seo et al. Continuous Control with Coarse-to-fine Reinforcement Learning. CoRL.
>
> ## 4. Clarification on Target Value Computation
>
> Specifically, we train two separate value networks and maintain two corresponding target networks, which are updated
> using exponential moving averages of the online value networks. The value targets are computed as the minimum of the two
> target network outputs.
>
> We will clarify this in the revised paper.
>
> ## 5. Improved Figure Readability
>
> We thank the reviewer for highlighting the readability issues in some figures. We will improve this in the revised
> paper.
>
> We greatly appreciate the reviewer's insightful feedback, which has helped us significantly improve the manuscript. We
> would be happy to incorporate any additional suggestions from the reviewer.

---

### Decision · Program_Chairs · 2025-05-01

**Decision:**

Accept (poster)

**Comment:**

This paper addresses the problem of reinforcement learning for continuous control with action discretization, specifically the explosion in dimensionality that occurs if the continuous action space is jointly binned in a grid-like manner across all dimensions. One solution that prior work has explored is the use of separate action binning across each dimension, which avoids the aforementioned explosion but in turn makes an assumption about action dimension independence. This work proposes a coarse-to-fine auto-regressive framework that overcomes both of these two issues.

Reviewers agree that the claims in the paper are supported by empirical evidence  and the proposed method is well-motivated. There were initial concerns about a lack of baselines and number of seeds used, but the authors were able to produce additional results during the rebuttal period that assuaged concerns. There were also concerns about assumptions for the theoretical contribution which were later clarified. If these changes are incorporated into the camera ready, this paper is a strong contribution to ICML.